

# Impacts of aerosol-radiation and aerosol-cloud interactions on a short-term heavy rainfall event - A case study in the Guanzhong Basin, China

Naifang Bei[1], Bo Xiao[1, 3], Ruonan Wang[2], Yuning Yang[1], Lang Liu[4], Yongming Han[1, 2], and
Guohui Li[2*]

[1]School of Human Settlements and Civil Engineering, Xi'an Jiaotong University, Xi'an, 710049, China
[2]Key Lab of Aerosol Chemistry and Physics, State Key Laboratory of Loess Science, Institute of Earth
 Environment, Chinese Academy of Sciences, Xi'an 710061, China
[3]Xi'an Meteorological Bureau, Xi'an, 710600, China
[4]College of Meteorology and Oceanography, National University of Defense Technology, Changsha, 410073,
 China

*Correspondence to*: Guohui Li (ligh@ieecas.cn)

**Abstract.** Atmospheric aerosols influence clouds and precipitation by aerosol-radiation interactions (ARIs) and
aerosol-cloud interactions (ACIs). In the study, the synergetic effect of ARIs and ACIs on development and
precipitation for a mesoscale convective system (MCS) occurred in the Guanzhong Basin (GZB) of central China
has been examined using a cloud-resolving fully-coupled Weather Research and Forecasting model with
chemistry (WRF-Chem). The model reasonably reproduces the temporal variation and spatial distribution of air
pollutants, the hourly rain rate and daily precipitation distribution against observations in the GZB. Sensitivity
simulations are conducted under different aerosol scenarios by adjusting the anthropogenic emissions. When the
ARI effect is not considered, the daily precipitation does not show an increasing trend with increasing aerosols in
the GZB, which is mainly caused by competition among convective clouds to available water vapor in
development of the MCS. ARIs exert two opposite effects on convection: stabilizing effect to suppress convection
and lifting effect to foster convection, which counteract each other. When the lifting effect outweighs stabilizing
effect, the updraft is enhanced, which increases precipitation in the GZB. However, the synergetic effect of ARIs
and ACIs significantly suppress precipitation when the PM pollution is severe. It is worth noting that the
synergetic effect consistently decreases the precipitation in the whole domain with increasing aerosols, but ARIs
play a more important role in the decreasing trend of the precipitation with deterioration of PM pollution.



## 1    Introduction


Atmospheric aerosols influence cloud and precipitation, including their initiation time, intensity, occurrence frequency, precipitation amount precipitation area, which currently represent the primary uncertainty in climate change drivers and future scenarios (IPCC, 2013). The aerosol effect on cloud and precipitation is primarily focused on two critical aspects, including the ability of aerosols to act as cloud condensation nuclei (CCN) and/or

ice nuclei (IN) and their influence on the atmospheric thermodynamic structure (Boucher et al., 2013; Huang and Ding, 2021; Zhao et al., 2024).

By serving as CCN and IN, aerosols can increase cloud albedo (Twomey, 1977), cloud longwave emissivity (Garrett and Zhao, 2006), and extend cloud lifetime (Albrecht, 1989; Christensen et al., 2020), which is referred to as aerosol indirect effect (AIE) or aerosol cloud interaction (ACI). By absorbing and scattering incident solar

radiation, aerosols can enhance atmospheric stability. This occurs as they cool the ground surface while heating the atmosphere, subsequently impacting the development of clouds and precipitation (Charlson et al., 1992; Sun and Zhao, 2021), which is also termed the aerosol direct effect (ADE). The absorbing aerosols within clouds can also accelerate evaporation of cloud droplets by absorbing solar radiation, leading to reductions in cloud cover and precipitation, and decreases in cloud albedo (Ackerman et al., 2000), which is named as semi-direct radiative

effect. Together with ADE, it is defined as aerosol-radiation interaction (ARI).

Despite significant advancements in understanding the mechanisms of ACIs, there are still considerable uncertainties regarding the impacts of aerosols on precipitation for various cloud regimes and environment conditions, especially in mixed-phase convective clouds (Tao et al. 2012; Wang et al., 2013; Li et al., 2019). The impact of ACIs on precipitation varies under different meteorological conditions (Khain et al., 2008; Storer et al.,

2010; Lebo and Morrison, 2014; Guo et al., 2016; Chen et al, 2020), cloud types (Tao, 2007; Lee et al., 2008), precipitation types (Guo et al., 2018; Sun and Zhao, 2021), cloud/precipitation development stages (Guo et al., 2014), aerosol composition and size distribution (Zhang et al., 2002; Jiang et al., 2018; Xi et al., 2024), and orography conditions (Yang et al., 2014; Nugent et al., 2016).

It has been well established that elevated aerosol concentrations increase the cloud droplet number

concentration (CDNC), thus reducing cloud particle sizes, inhibiting collision and coalescence processes and increasing the cloud liquid. Increased cloud liquid in the air is evaporated or frozen through enhancing freezing of cloud liquid and parcel buoyancy. The increased cloud liquid mass can further invigorate convection and enhance precipitation (Rosenfeld et al., 2008; Chen et al., 2017; Dagan et al., 2017). Furthermore, increased cloud liquid mass can increase evaporation to strengthen gust fronts, which reinforces convective clouds and the related

precipitation amount in turn (Khain et al., 2005; Tao, 2007; Lee et al., 2018). Aerosols have also been demonstrated to suppress the precipitation of warm clouds across various regions globally (Ackerman et al., 2003; Wang et al., 2011). Decreased droplet sizes within aerosol-laden clouds restrain the coalescence of droplets, which can slow the conversion from cloud droplets into rainwater, to the extent of completely suppressing the formation of warm rain (Rosenfeld et al., 2001).

Lau et al. (2008) have found that aerosols broadly influence atmospheric conditions by affecting the thermodynamic properties and modifying large-scale circulation through various feedback processes. However, the radiative impacts of absorbing versus scattering aerosols differ significantly, leading to diverse effects on cloud formation and precipitation (Li et al. 2017; Sun and Zhao, 2021). Absorbing aerosols can significantly alter



the development of clouds and precipitation by changing the vertical temperature profiles and disturbing local
atmospheric circulation (Sun and Zhao, 2021). Their impacts on precipitation are especially related to the relative
position of aerosols with respect to clouds (Kaufman et al., 2006; Wilcox et al., 2012). While scattering aerosols
tend to cool the ground surface, thereby enhancing atmospheric stability and inhibiting both convection and
rainfall. Due to the complex and nonlinear nature of aerosol radiative interactions with cloud-precipitation
processes, variations in aerosol levels can lead to a shift from enhancing precipitation to suppressing it (Jiang et
al. 2016; Wang et al., 2023). ARIs can also impact precipitation by altering wind speed, which primarily due to
reduced water vapor advection and evaporation (Koren et al., 2004; Yang et al., 2013a).

Fast-growing industries and city expansions have substantially increased aerosol levels over the past 3
decades in Guanzhong Basin (GZB) of central China (Bei et al., 2016a; 2016b; 2017b). The basin is located in
the transitional zone between the Qinling Mountains and the Loess Plateau (34°~35.5°N, 106.5°~110.5°E, please
refer to Figure 1: the area surrounded by blue lines). Studies on the aerosol impact on precipitation in the GZB
and surrounding areas (GZBs) are mostly focused on the rainfall over the mountain area (Rosenfeld et al., 2007;
Yang et al., 2013a; 2013b). Rosenfeld et al. (2007) have proposed that the hilly precipitation at Mt. Hua near
Xi'an could be decreased by 30 to 50% during hazy conditions. Yang et al. (2013b) have found that the decreasing
trend of orographic precipitation are correlated well with deterioration of the air pollution at Mt. Hua and in the
GZB based on more observational analyses, supporting for the hypothesis that both aerosol microphysical and
radiative effects could reduce orographic precipitation.

In the present study, we examine the synergetic effects of ARIs and ACIs on a short-term heavy precipitation
event occurred in the GZB using a fully coupled cloud-resolving WRF-Chem model. The WRF-Chem model and
experiment design are described in Section 2. Section 3 presents results and discussion. A summary and
conclusions are given in Section 4.

## 2    WRF-Chem model and experiments design

### 2.1    WRF-Chem Model

A specific version of the WRF-Chem model, with modifications by Li et al. (2010; 2011a; 2011b; 2012)
based on the original version by Grell et al. (2005), is used to study impacts of anthropogenic aerosols on a short-
time heavy rainfall event occurred in the GZB on July 24, 2016. Goddard shortwave module developed by Chou
and Suarez (1999) and Chou et al. (2001) is employed to account for the ARI effect. A two-moment bulk
microphysics scheme with aerosol effects developed by Morrison et al. (2009) is used to consider the ACI effect.
Detailed model description of the WRF-Chem model, the calculation of aerosol optical properties, and activation
of aerosols to CCN and IN can be found in Supplement Information (SI, S1, S2, and S3).

### 2.2    Experiments design

In this study, the WRF-Chem model is configured with two one-way nested grids with spacing of 9 km
(301×301 grid points) for domain 1 (D01) and 3 km (301×301 grid points) for domain 2 (D02). The two domains
are both centered at Xi'an (34.25°N, 109°E) (Figure 1). The simulations of D01 provide meteorological and
chemical initial and boundary conditions for D02. The simulations of D02 is primarily used to investigate the
impact of ACIs and ARIs on precipitation in the GZB. The vertical dimension is divided into 51 layers, extending



from the ground level up to the 50 hPa altitude. The vertical grid spacing is designed with a stretch, starting at 30 m near the ground surface and increasing to 400 m above 2.5 km. This configuration aims to enhance the resolution within the planetary boundary layer (PBL), thereby capturing finer details in this critical atmospheric region.

The WRF-Chem is first integrated for a 60-h period from 1200 UTC 21 to 0000 UTC 25, July 2016 for D01. The meteorological initial and boundary conditions are from the National Centers for Environmental Prediction (NCEP) final operational global gridded analysis (FNL) (1°×1°). The chemical initial and boundary conditions are derived through interpolation from the 6-h output of a global chemical transport model for $O_3$ and related chemical tracers (MOZART) (Horowitz et al., 2003). We vary the aerosol concentrations in the atmosphere

through adjusting anthropogenic emissions. A set of 41 anthropogenic emission scale factor (AESF) is used in numerical experiments, ranging from $2^{-3}$ to $2^3$ with an exponential increasing step of 0.15. Total 41 sensitivity simulations are conducted for D01.

   The WRF-Chem model is then integrated for a 24-h period from 0000 UTC 24 to 0000 UTC 25, July 2016 for D02. In order to investigate the impacts of aerosols with different concentrations on the short time heavy

rainfall event through ARIs, ACIs, and both of them, two groups of experiments are designed based on D02. Both ARI and ACI effects are considered in the benchmark simulation, in which the meteorological and chemical initial and boundary conditions are interpolated from the simulation of D01 with the AESF of 1.0 and the AESF is set to 1.0 (hereafter referred to as CTRL). The results in the CTRL are used to validate the model performance. Based on the CTRL, first group of sensitivity experiments are conducted, in which the AESF is adjusted according to

that for D01 (hereafter referred as to F_BASE). The chemical initial and boundary conditions for the member of F_BASE are interpolated from the corresponding member of D01 with the same AESF. The second group of sensitivity experiments is the same as the F_BASE but the ARI effect is turned off (hereafter referred as to F_ARI0). The model setup is the same for all experiments, except for the anthropogenic emission amplitude. Detailed model configuration can be found in Table S1.

**2.3  Model validation and statistical metrics**

   Hourly precipitation observations at meteorological sites with rain gauge in the GZBs are from China Meteorological Administration. Hourly observations of air pollutants, including $PM_{2.5}$, $O_3$, $NO_2$, and $SO_2$ are from the Ministry of Ecology and Environment of China. The performance of the WRF-Chem model simulations is assessed by comparing them with observations using metrics including the mean bias (*MB*), root mean square

error (*RMSE*), and the index of agreement (*IOA*). The population mean (*p-mean* hereinafter) of a given variable across all qualified grid points is used to assess the general impact of aerosol variations on cloud or cloud systems. Detailed description about *MB*, *RMSE*, *IOA* and *p-mean* can be found in Text S4.

**3  Results and discussion**

**3.1  Case descriptions and model validation**

The selected heavy rainfall event occurred on July 24-25, 2016 in the GZB. The rain gauge observations show that the 24-hr accumulated rainfall reaches 100 mm over the GZB with the maximum hourly precipitation



of 66.6mm occurring in Xi'an. During the event, the GZB is located near the bottom of a trough at 850 hPa, in the front of the trough at 700hPa and 500hPa, and in the center of a high pressure system in the upper level (200hPa), which are basically conducive to occurrence and development of low-level convections (Figure S1).

The radiosonde observations on 24 and 25 July are shown in Figure S2. The maximum convective available potential energy (CAPE) of 5045 J kg$^{-1}$ is observed at 1200 UTC 24 July (Figure S2a), which is right before the heavy rainfall peak (1400 UTC) in Xi'an. The CAPE decreases significantly after the heavy rainfall event (Figures S2c).

Figure 2 presents the spatial distribution of simulated and observed concentrations of PM$_{2.5}$, O$_3$, NO$_2$, and

SO$_2$ at 0000 UTC on 24 July 2016. In general, the model reasonably simulates the pattern of air pollutants compared to observations at monitoring sites in the GZB. The PM$_{2.5}$ concentration generally exceeds 50 ug m$^{-3}$, and the SO$_2$ level is also high at urban areas and industrial zones in the basin. Figure 3 shows simulated (red line) and observed (black dots) diurnal profiles of hourly mass concentrations of PM$_{2.5}$, O$_3$, NO$_2$, and SO$_2$ averaged at monitoring sites in the GZB on 24 July 2016. The model yields the reasonable temporal variations of air pollutants

against observations, particularly regarding to PM$_{2.5}$ and O$_3$, with the IOA of 0.88 and 0.96, respectively. The model tends to underestimate the PM$_{2.5}$ concentration during daytime, with a MB of -4.5 μg m$^{-3}$.

Figure 4 shows time series of precipitation rates averaged at meteorological sites with rain gauge in the GZB and GZBs on 24 July 2016. The model performs well in simulating the hourly rain rate compared to observations. For example, the enhancement of rain rate from 1000 to 1400 UTC is reproduced, and the rapid falloff from 1400

to 1800 UTC is simulated. The MB and IOA are -0.03 mm h$^{-1}$ and 0.98 in the GZB, and -0.01 mm h$^{-1}$ and 0.96 in the GZBs, respectively. Figure 5 presents the pattern comparison of the daily precipitation in the GZBs. The model generally replicates the precipitation distribution against the observations, for instance, the maximal precipitation center in the central GZB is well simulated. However, there exist considerable underestimation and overestimation of precipitation, showing difficulties in simulating convective rainfall with the model.

**3.2 Impacts of ARIs on meteorological fields in the GZB**

We first examine ARI effects on the profile of temperature and water vapor in the morning (from 0000 to 0400 UTC), since after 0400 UTC clouds commence to form and develop, with occurrence of sporadic precipitation in the GZB. The cloud optical thickness (COT) is far greater than the aerosol optical depth (AOD), so the thermodynamic effect of cloud perturbation caused by ACIs would conceal that of ARIs.

Figure 6a shows the variation of the average near-surface PM$_{2.5}$ concentration in the morning in the GZB with increasing anthropogenic emissions. Near-surface PM$_{2.5}$ concentrations monotonically increase with increasing anthropogenic emissions as expected, but the relationship is nonlinear. When the AESF increases from 0.125 to 1.0, near-surface [PM$_{2.5}$] increase by 6.6 times. However, when the AESF increases from 1.0 to 8.0, the enhancement of [PM$_{2.5}$] is 9.4 times, which is mainly caused by the ARI effect which suppresses development of

the PBL to increase near-surface air pollutants level and the enhanced formation of secondary aerosols (Wu et al., 2019). As the major absorbing aerosol in the atmosphere, the near-surface concentration of black carbon also increases with increasing AESF, but its linear relationship with the AESF is better than that of PM$_{2.5}$ (Figure 6b). The AOD and absorbing AOD (AAOD) also reveal the similar monotonically increasing relationship with the AESF in the GZB (Figures 6c-d). When the AESF is 1.0 (CNTL case), the AOD and AAOD are about 0.44 and



0.04, with the single scattering albedo of about 0.91, indicating a moderately strong absorbing atmosphere over the GZB.

Aerosols in the atmosphere attenuate incident solar radiation by scattering and absorption and further decrease the solar radiation down to the surface, causing less sensible heat flux to lower the temperature of low-level atmosphere. However, aerosol light absorption tends to heat the atmosphere. Figure 7a provides the ARI

effect on the average temperature profile from 0000 to 0400 UTC over the GZB by comparing the F_BASE and F_ARI0 under different aerosol conditions. The ARI effect lowers the temperature of the low-level atmosphere, and the temperature decrease becomes increasingly significant with increasing AESF, but insignificant as height increases. Absorbing aerosols heat the atmosphere from around 900~1000m to 2000~2500m and the thickness of the heated atmosphere increases with increasing AESF. Previous studies have also reported such a phenomenon

(Ding et al., 2016; Gao et al., 2016; Wilcox et al., 2016). Meanwhile, the perturbation of temperature profile caused by ARIs also suppresses development of the PBL, which does not facilitate dispersion of air pollutants and water vapor in the PBL. The ARI effect increases the mass mixing ratio of water vapor in the atmosphere below around 500m and decreases it in the atmosphere from about 500m to 1700m (Figure 7b). Therefore, the ARI effect increases the atmospheric stability, which tends to inhibit cloud formation and development. However,

the temperature enhancement caused by absorbing aerosols above the PBL cause a "warm bubble" effect, which could induce updrafts to promote convection. As shown in Figure 7c, the heating effect of ARIs generates a secondary upward movement in the atmosphere above around 300m. Interestingly, ARIs exert two opposite effects on cloud formation and development, i.e. stabilizing effect and lifting effect, which counteract each other. If stabilizing effect outweighs lifting effect, ARIs inhibit cloud formation and development, and it is opposite

when lifting effect outweighs stabilizing effect.

### 3.3    Response of cloud properties to changes of aerosols

We then investigate the effect of ACIs and ARIs on cloud properties and precipitation during the main precipitation period from 0800 to 1800 UTC. Figure 8a presents the dependence of the p-mean of CDNC over the GZB from 0800 to 1800 UTC on the AESF, revealing an increasing of CDNC with increasing AESF in F_BASE

and F_ARI0. Increase anthropogenic emissions increase aerosol concentrations, providing more CCN to activate to form cloud droplets, which has been reported in many previous studies (Li et al., 2008; 2009). The ARI effect considerably influences the CDNC with the same AESF (Figure 8b). When the AESF is less than 0.33, the ARI effect decreases the CDNC. With the AESF exceeding 0.33, the ARI effect increases the CDNC, and the enhancement of the CDNC becomes increasingly significant with the AESF exceeding 1.6.

It has been well established that elevated aerosols increase CDNC and decrease the droplet size, inhibiting collision and coalescence processes and further leading to more cloud water in the air. In the F_ARI0, the cloud water path (CWP) averaged in the GZB from 0800 to 1800 UTC generally increases with the AESF when the AESF is less than 5.4 (Figure 8c). However, in the F_BASE, the CWP shows an increasing trend with the AESF when the AESF is less than 1.6, and when the AESF exceeds 1.6, the CWP fluctuates in the range between 50

and 56 g m$^{-2}$. In addition, only when the AESF is in the range between 0.27 and 0.70, the ARI effect increases the CWP (Figure 8d). When the AESF exceeds 1.6, the ARI effect decreases the CWP by more than 10%. Apparently, the ARI effect on the CDNC and CWP varies with the AESF. When the AESF is in the range between 0.33 and



0.70, the ARI effect simultaneously increases the CDNC and CWP. With the AESF exceeding 0.70, the ARI effect increases the CDNC but decreases CWP.

The ARI effect can be reflected by variations of the updraft in the GZB. Figure 9a shows the variation of the updraft averaged from 0800 to 1800 UTC over the GZB with the AESF. In the F_ARI0, increasing AESF or aerosols does not enhance updrafts or even slightly weaken the updrafts, particularly with the AESF exceeding 2.3. Previous studies have proposed that increased aerosols reduce cloud particle sizes to decrease the efficiency of collision and collection, increasing the freezing of cloud droplets and associated latent heat release above the

0°C isotherm and invigorating convective clouds (Li et al., 2008; Rosenfeld et al., 2008; Chen et al., 2017; Dagan et al., 2017). However, in the study, increased aerosols do not increase the ice-phase hydrometeors which generally show a slight decreasing trend with the AESF in F_ARI0 (Figure 9c). The main reason for decreasing trend of updrafts with increasing aerosols is limitation of available water vapor in the development of the mesoscale convective system (MCS). If some convective clouds in a MCS are invigorated by increased aerosols,

more cloud water and ice-phase hydrometeors are produced in those clouds, which decreases the available water vapor for other cloud development. With increasing AESF or aerosols, the formation of ice-phase hydrometeors are gradually inhibited due to increasing small cloud droplets, causing decrease of updrafts (Figure 9d).

The ARI effect enhances the updraft in the GZB with the AESF of 0.125 and in the range between 0.4 and 1.6, and weakens it under other AESF conditions (Figure 9b). As discussed above, ARIs cause the stabilizing and

lifting effect on convections. When the lifting effect surpasses the stabilizing effect, updrafts are intensified and convection is fostered by ARIs. Menon et al. (2002) have reported that absorbing aerosols over Asia can increase low-level convergence and vertical velocity, overcoming the stabilizing effects of ARIs to enhance the summer monsoonal circulation. Li et al. (2016) have demonstrated that ARIs induced by absorbing aerosols could vary the thermodynamic stability and convective potentials of the low-level atmosphere, reinforcing the early East

Asian summer monsoon. However, when the stabilizing effect outweighs the lifting effect, the convective available potential energy (CAPE) is reduced by ARIs and the upward movement is suppressed. When the AESF is more than 2.0, the ARI effect decreases the updraft by more than 6% and the updraft decrease generally becomes increasingly significant with increasing AESF.

It is worth noting that the variation of CWP caused by ARIs is well correlated with that of updraft in the

GZB, with a correlation coefficient of 0.87, showing that ARI induced enhancement of updrafts favors water vapor condensation on cloud droplets. However, the variation direction of the two variables is not always consistent. The variation of CDNC due to ARIs shows a negative correlation with that of updraft, with a coefficient of -0.86. Generally, increased updraft elevates water vapor supersaturation to activate more aerosols, increasing CDNC and further enhancing water vapor condensation. The CDNC is also determined by the conversion

efficiency from cloud to rain water, which is dependent on the effective radius of cloud droplets. Figure 10a illustrates the dependence of p-mean of cloud effective radius ($R_{eff}$) on the AESF in F_BASE and F_ARI0. $R_{eff}$ decreases monotonically with increasing AESF in the two group simulations. In general, increasing $R_{eff}$ decreases the autoconversion from cloud to rain water. With the AESF exceeding 0.24, the ARI effect decreases $R_{eff}$ and the $R_{eff}$ is reduced by over 5% when the AESF is more than 1.6 (Figure 10b). The variation of CDNC due to ARIs is

highly correlated with that of $R_{eff}$, with a correlation coefficient of about 0.98.

### 3.4    Aerosol effects on precipitation



Figure 11a provides the variation of the average daily precipitation in the GZB with the AESF in F_BASE and F_ARI0. When ARIs are excluded or only ACIs are considered in F_ARI0, the daily precipitation shows a nonlinear relationship with the AESF. When the AESF is less than about 2.4, the precipitation is not sensitive to the AESF, and does not show increasing or decreasing trend with the AESF, fluctuating in the range between 15.0 and 16.0 mm d$^{-1}$. With the AESF exceeding 2.4, the precipitation generally shows a decreasing trend with the AESF. In the F_BASE with the ARI effect, the precipitation also shows fluctuation with the AESF less than 2.4, but the fluctuation amplitude is more significant than that in F_ARI0, in the range between 14.0 and 17.0 mm d$^{-1}$. When the AESF is more than 2.4, the precipitation decreases rapidly with increasing AESF.

Multifarious measurements and numerous modeling simulations have revealed that increased aerosols invigorate convective clouds and enhance precipitation (Cerveny and Balling, 1998; Shepherd and Burian, 2003; Khain et al., 2005; Lin et al., 2006; Tao, 2007; Li et al., 2008; Lee et al., 2018). However, we do not observe significant increasing trend of precipitation with increasing AESF in both F_BASE and F_ARI0. In addition, when the AESF is more than 1.0, the decreasing trend of precipitation with increasing AESF becomes significant. Elevated aerosols increase CDNC and cloud water content and reduces droplet size to inhibit autoconversion, enhancing glacier processes to invigorate convection and further precipitation. However, when the droplet size is decreased to a threshold due to increased aerosols, the glacier process is inhibited and convection commences to be weakened, causing decrease of precipitation. Although our simulations show that CDNC and CWP increase monotonically with increasing AESF, the total ice-phase hydrometeors and updrafts do not have significant increasing trend with the AESF less than 1.0. When the AESF is more than 1.0, those two variables exhibit decreasing trend with the AESF. As discussed above, the main reason is competition between convective clouds to available water vapor in the MCS.

The precipitation variation caused by the ARI effect fluctuates in the range between -7.0% and 7.0% with the AESF less than 1.7 (Figure 11b). When the AESF is more than 1.7, the ARI effect substantially decreases precipitation. The precipitation variation due to ARIs is highly correlated with that of updraft, with a coefficient of 0.94. In addition, the variation direction of the two variables is generally the same, showing that the ARI induced variation of updraft is the main reason for the precipitation variation.

Figure 12 shows the daily precipitation distribution under different AESF in the F_ARI0. The maximal precipitation center is generally concentrated in the central GZB, but the distribution of heavy and torrential rainfall presents significant changes with increasing AESF. The impact of ARIs on precipitation distribution in the GZB is not very significant when the AESF is less than 1.0. With the AESF exceeding 1.0, the ARI effect significantly decreases the area with heavy and torrential rainfall, particularly with regarding the torrential rainfall (Figure 13). As shown in Figure 9b, the ARI effect increases the total precipitation in the GZB by about 6.5% with the AESF of 0.35. Comparing Figure 12c and Figure 13c, the ARI effect also increases considerably the area with heavy rainfall, but the precipitation in the south of the GZB is decreased. The lifting effect due to ARIs in the GZB increases the vertical wind speed and induce the horizontal convergence above the PBL, causing the transport of water vapor from its surroundings, which enhances precipitation in the GZB but reduces precipitation in its surrounding. Comparing Figure 12h and Figure 13h, the stabilizing effect due to ARI decreases the precipitation in the GZB but increases it in the south of the GZB.

We further investigate the impact of ACIs and ARIs on the precipitation in the whole domain in which the water vapor input and output are fixed in lateral boundaries. Therefore, the total water vapor mass is conserved



or not altered by the aerosol effect in the whole domain during the integration period. In the F_ARI0, when the AESF is less than 1.7, the precipitation varies insignificantly with the AESF, i.e., the mean of the precipitation for the first 26 members is 8.15 mm d$^{-1}$, but the standard deviation is only 0.01 m d$^{-1}$ (Figure 14a). With the AESF

exceeding 1.7, the precipitation shows a decreasing trend with the AESF. When the ARI effect is considered in the F_BASE, increasing aerosols consistently decreases the precipitation of the whole domain. The ARI effect also consistently decreases the precipitation with the same AESF, and the precipitation decrease is more than 10% when the AESF exceeds 1.0 (Figure 14b). Additionally, the lifting effect induced by ARIs also considerably modulates the decreasing trend of precipitation with the AESF. The variation of p-mean of the updraft due to

ARIs in the domain is highly correlated with that of the precipitation, with the correlation coefficient of 0.96 (Figures 14c, d).

## 4 Summary and conclusions

A MCS occurred in the GZBs with heavy rainfall on 24 July 2016 has been investigated using a fully coupled cloud-resolving WRF-Chem model. The synergetic effect of ACIs and ARIs on the precipitation process

of the MCS has been assessed by sensitivity studies with various aerosol scenarios through adjusting the anthropogenic emissions.

The WRF-Chem model generally well replicates the temporal variation and spatial distribution of air pollutants when comparing to measurements in the GZB. The model also performs well in simulating the hourly rain rate and reasonably reproduces the daily precipitation against observations in the GZB and GZBs.

Sensitivity simulations show that ARIs generally cools the atmosphere near the ground surface but heat it above the PBL, causing the stabilizing and lifting effect which exert opposite impacts on convection. When ARIs are not considered, the daily precipitation in the GZB is not sensitive to aerosol concentrations when the particulate pollution (PM) is not severe, but shows a decreasing trend with further deterioration of PM pollution. The main reason for non-increasing trend of the precipitation with increasing aerosols is competition among convective

clouds to available water vapor in development of the MCS. Too many small cloud droplets caused by increased aerosols also inhibit the formation of ice-phase hydrometeors to decrease updrafts and further the precipitation.

The ARI effect considerably modulates the precipitation in the GZB. When the lifting effect outweighs the stabilizing effect, the updraft is enhanced, causing the increase of the precipitation in the GZB. However, it is opposite when the stabilizing effect outweighs the lifting effect. However, the ARI effect does not enhance the

precipitation in the whole domain with the same anthropogenic emission. In addition, the synergetic effect of ACIs and ARIs consistently decreases the precipitation in the whole domain, but ARIs considerably modulate the decreasing trend of the precipitation.

**Acknowledgements.** This work is financially supported by the National Key Research and Development Program

of China (grant no. 2022YFF0802502), Naifang Bei acknowledges the National Natural Science Foundation of China (grant no. 41975175), and the Key Research and Development Program of Shaanxi (grant no. 2024SF-ZDCYL-05-05).

**Code and data availability.** The hourly ambient surface O$_3$, NO$_2$ and PM$_{2.5}$ mass concentrations are real-timely

released by Ministry of Environmental Protection, China on the website http://www.aqistudy.cn/, freely



downloaded from http://106.37.208.233:20035/ (China MEP, 2013). Precipitation observations at meteorological sites with rain gauge in the GZBs are from China Meteorological Administration, which can be accessed at https://data.cma.cn/data/cdcdetail/dataCode/A.0012.0001.html.

**Author contributions.** GL, as the corresponding author, provided the ideas and financial support, verified the conclusions, and revised the paper. NB conducted research, designed the experiments, performed the simulation, processed the data, prepared the data visualization, and prepared the manuscript, with contributions from all authors. XB and YH provided the data and primary data processing and reviewed the manuscript. RW validated the model performance, analyzed the study data, and reviewed the manuscript. YY and LL analyzed the initial

simulation data and visualized the model results.

**Competing interests.** The authors declare that they have no conflict of interest.

**Financial support.** This work is financially supported by the National Key Research and Development Program

of China (grant no. 2022YFF0802502), the National Natural Science Foundation of China (grant no. 41975175), and the Key Research and Development Program of Shaanxi (grant no. 2024SF-ZDCYL-05-05).



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



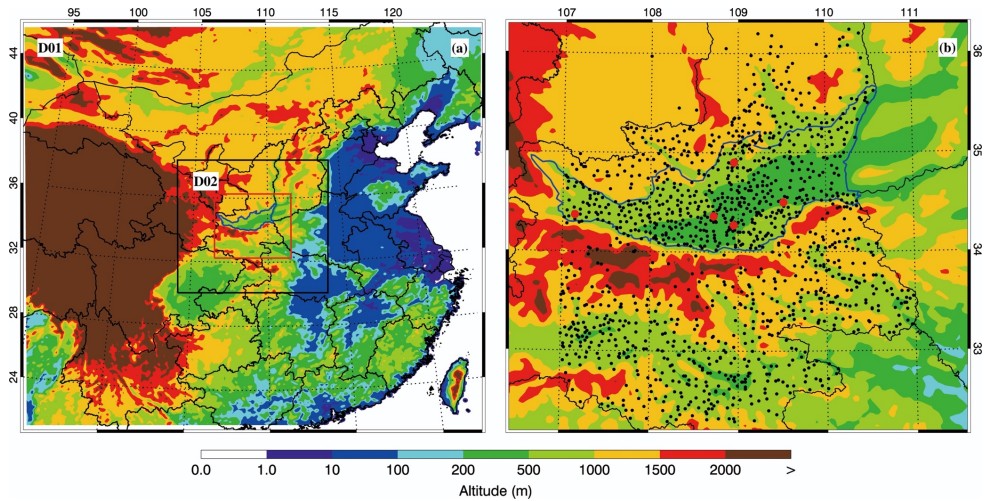

**Figure 1: (a) WRF-Chem simulation domain with topography and (b) Guanzhong basin with monitoring sites. In (a) and (b), the area surrounded by blue lines represents the Guanzhong basin. In (b), the black dots denote the meteorological sites with rain gauge and the red dots denote the sites with air pollutants observations.**



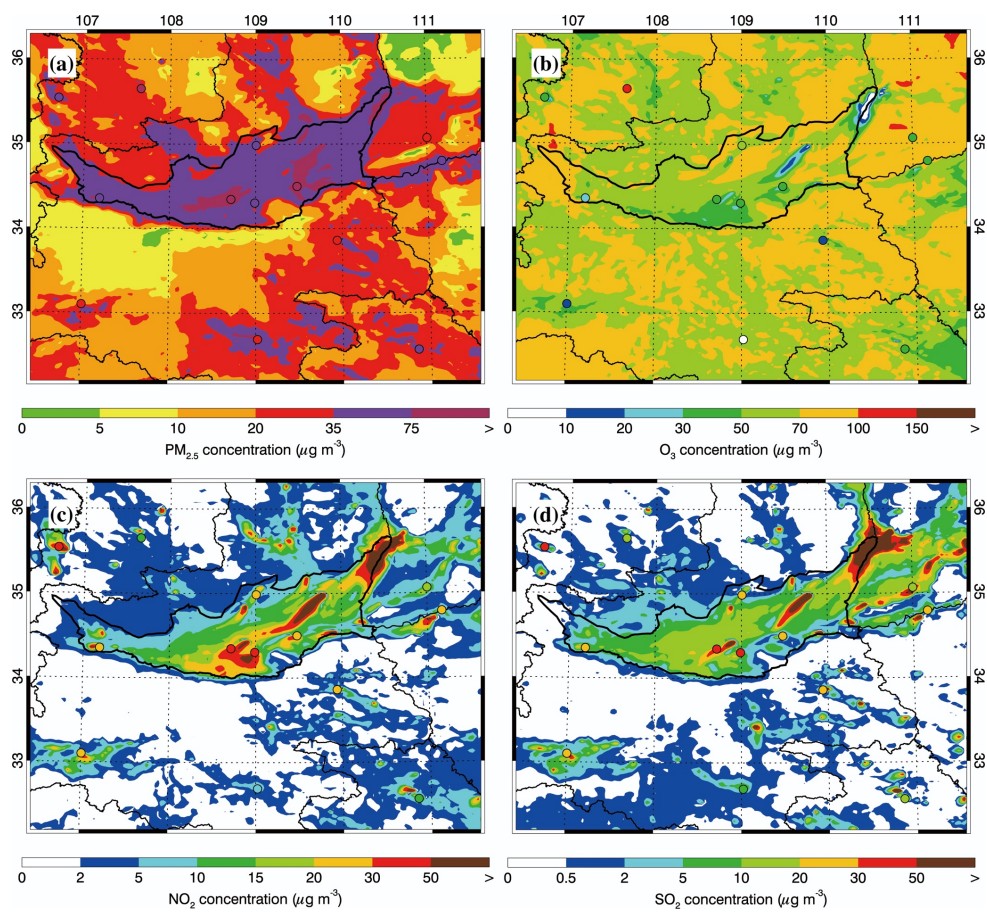


**Figure 2: Pattern comparisons of simulated (color counters) vs. observed (colored dots) near-surface mass concentrations of (a) PM$_{2.5}$, (b) O$_3$, (c) NO$_2$ and (d) SO$_2$ at 0000 UTC on 24 July 2016.**




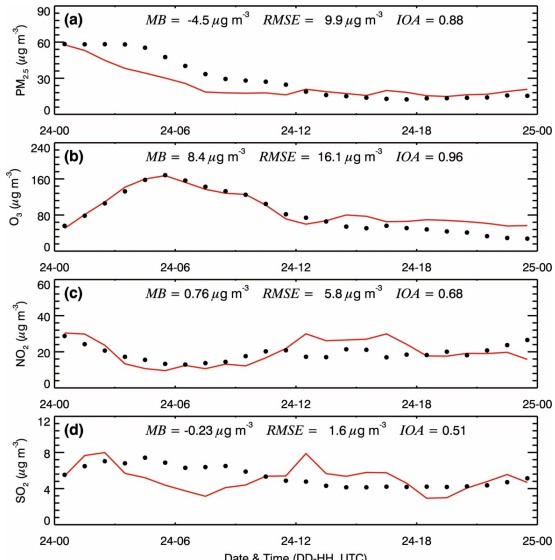


**Figure 3: Comparison of observed (black dots) and simulated (solid red lines) diurnal profile of near-surface hourly mass concentrations of (a) PM$_{2.5}$, (b) O$_3$, (c) NO$_2$, and (d) SO$_2$ averaged at monitoring sites in the GZB on 24 July 2016.**



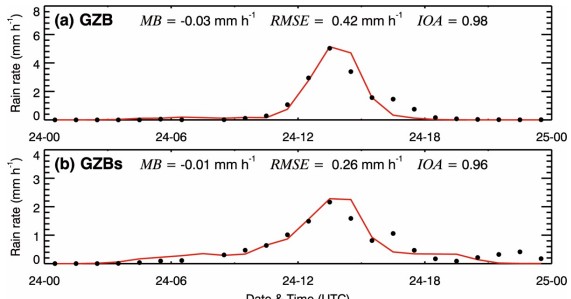


**Figure 4: Comparison of observed (black dots) and simulated (solid red lines) diurnal profile of hourly rain rate averaged at monitoring sites in the (a) GZB and (b) GZBs on 24 July 2016.**



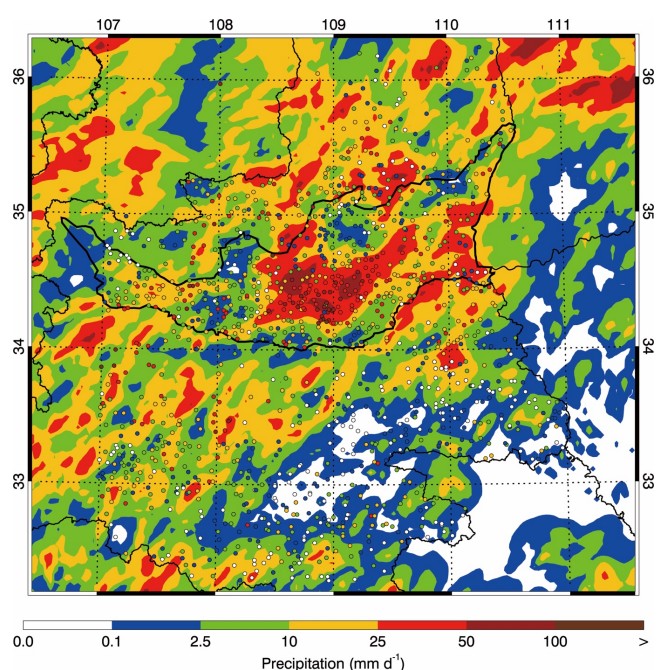


**Figure 5: Pattern comparisons of simulated (color counters) vs. observed (colored dots) accumulative precipitation on 24 July 2016.**



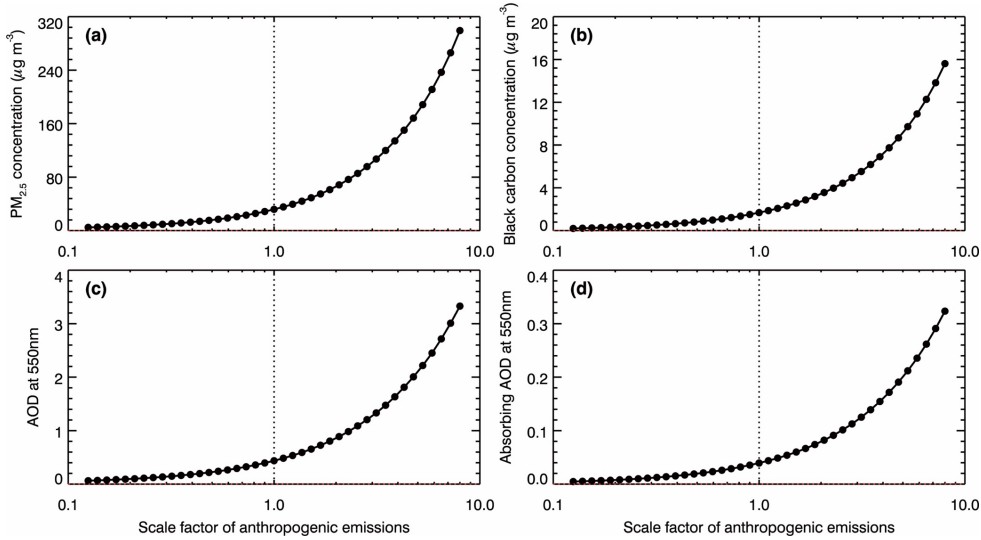


**Figure 6: Average (a) near-surface PM$_{2.5}$ (b) black carbon mass concentration, (c) AOD at 550nm, and (d) absorbing AOD at 550 nm in the GZB from 0000 to 0400 UTC, as a function of the scale factor of anthropogenic emissions.**



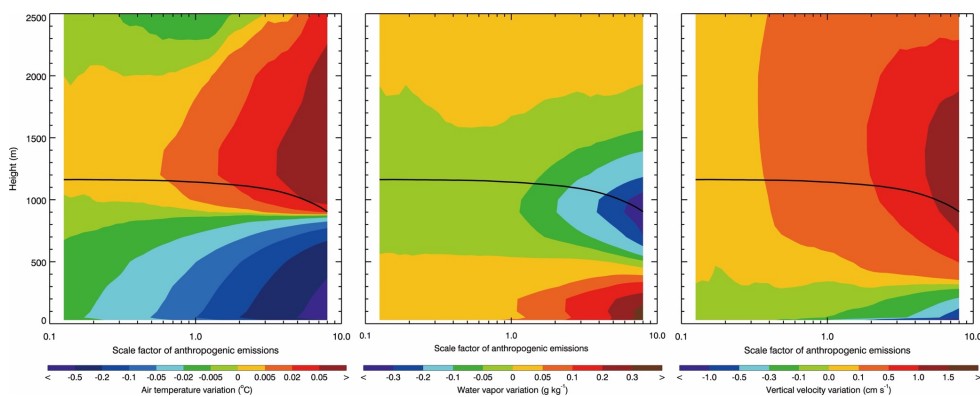


**Figure 7: Average profile variation of (a) air temperature, (b) water vapor, and (c) vertical velocity in the GZB from 0000 to 0400 UTC caused by ARIs, as a function of the scale factor of anthropogenic emissions. The black line denotes the PBL height.**



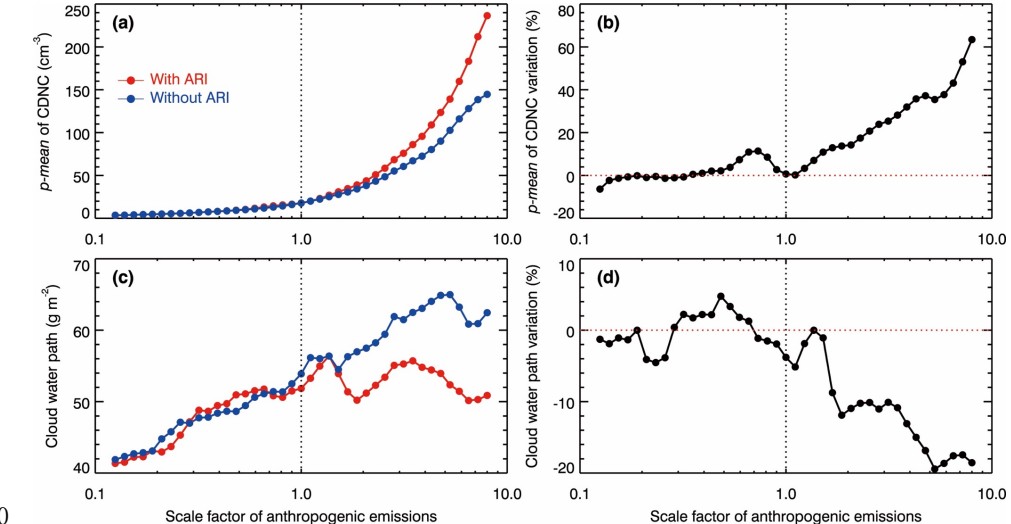


**Figure 8: (a) p-mean of CDNC, (b) variation of p-mean of CDNC due to ARIs, (c) average CWP, and (d) variation of CWP due to ARIs over the GZB from 0800 to 1800 UTC, as a function of the scale factor of anthropogenic emissions.**





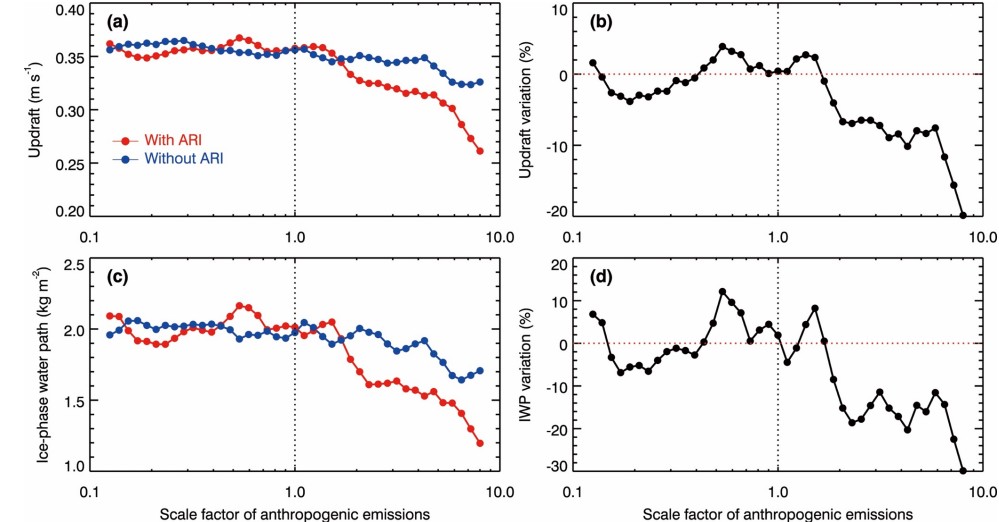

**Figure 9: Average (a) updraft, (b) variation of updraft due to ARIs, (c) IWP, and (d) variation of IWP due to ARIs over the GZB from 0800 to 1800 UTC, as a function of the scale factor of anthropogenic emissions.**






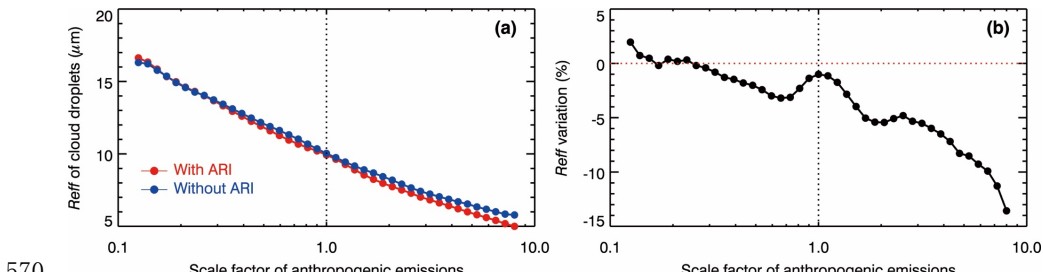


**Figure 10: Average (a) Reff of cloud droplets and (b) variation of Reff due to ARIs over the GZB from 0800 to 1800 UTC, as a function of the scale factor of anthropogenic emissions.**






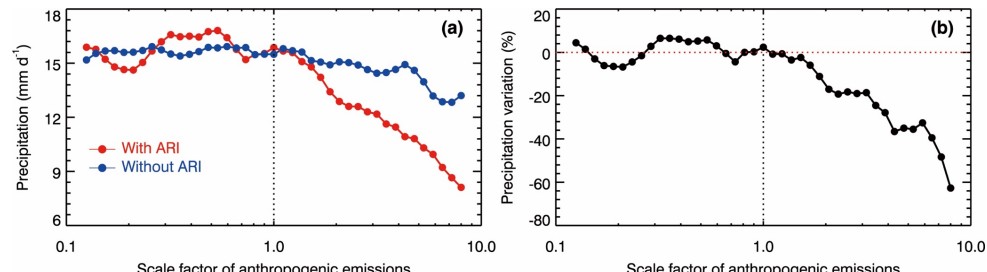

**Figure 11: Average (a) accumulative precipitation and (b) variation of precipitation due to ARIs in the GZB on 24 July 2016, as a function of the scale factor of anthropogenic emissions.**





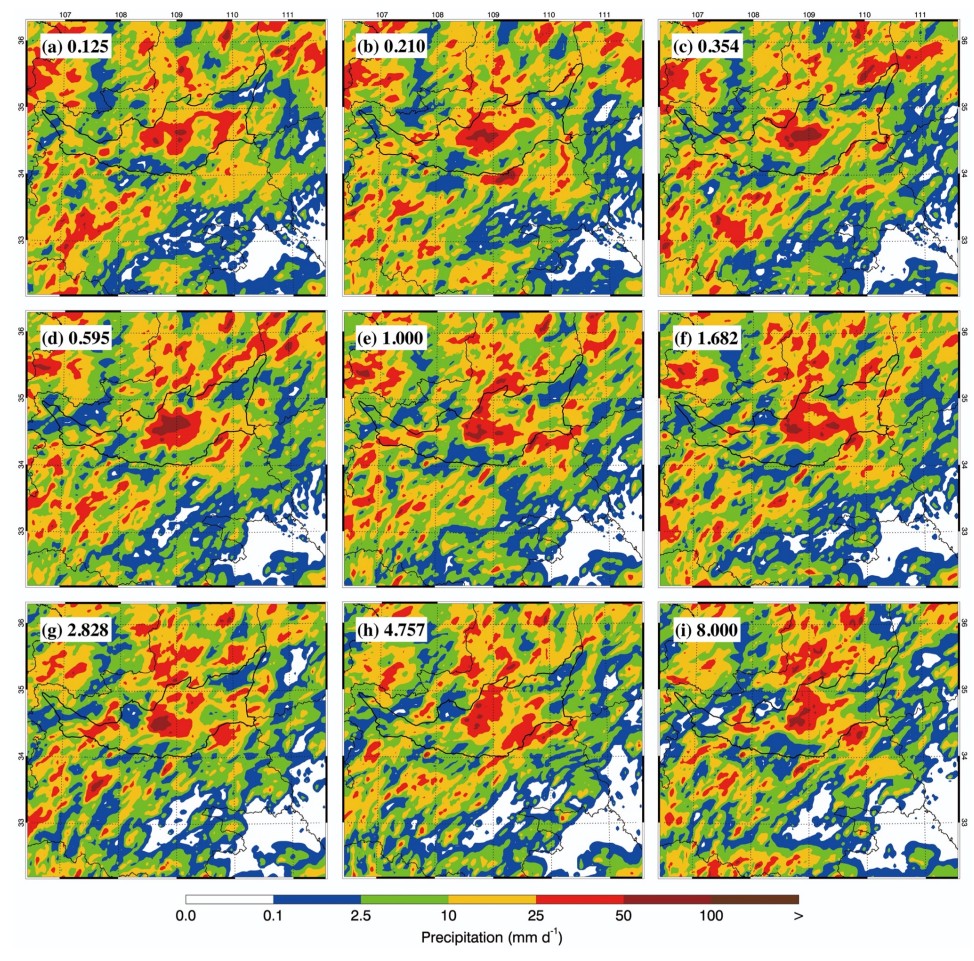

**Figure 12: Accumulative precipitation distribution on 24 July 2016 for various scale factor of anthropogenic emissions when ARIs are not considered.**



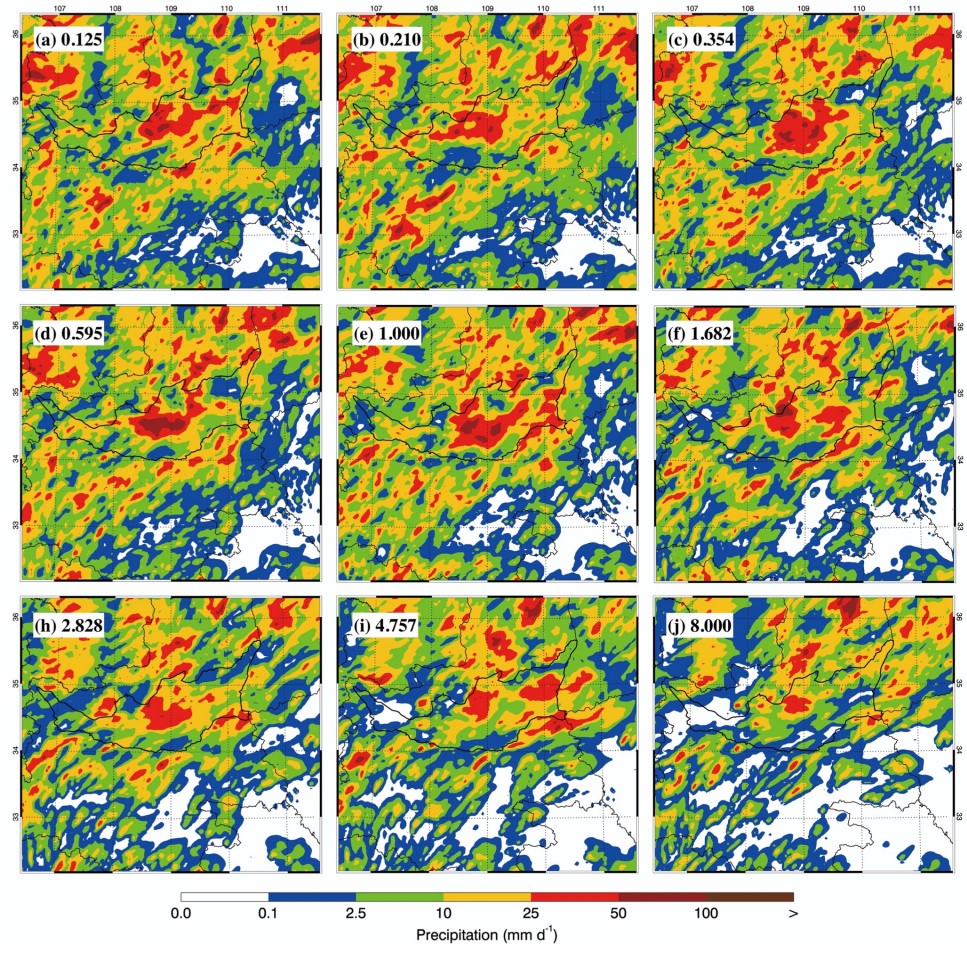

**Figure 13: Accumulative precipitation distribution on 24 July 2016 for various scale factor of anthropogenic emissions when ARIs are considered.**





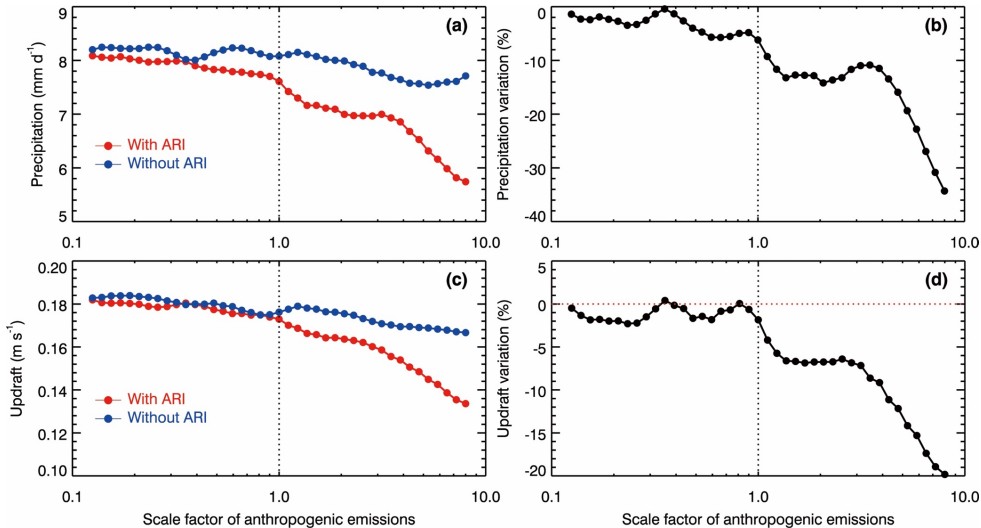

**Figure 14: Average (a) daily precipitation, and (b) variation of daily precipitation, (c) updraft and (d) variation of updraft due to ARIs from 0800 to 1800 UTC over the whole domain, as a function of the scale factor of anthropogenic emissions.**
