# Peer review of "Impacts of aerosol-radiation and aerosol-cloud interactions on a short-term heavy rainfall event - A case study in the Guanzhong Basin, China"

_EGUsphere, 2024_

## Referee Comment (RC1)

Review

[General Comments]

Time Zone: I was confused by time zones throughout the manuscript. When you are talking the results of model simulations, I know they should be based on UTC by default. But when you are talking about observations, it is little tricky because I don't know whether they are using the local time. Also, you may add the local time next to UTC time, otherwise it will be a confusion. For example, L160 mentioned morning time as well as from 0000 to 0400 UTC, the readers have to convert UTC time to local time to understand you are talking about from 0800 to 1200 local time. Also, you may add all local time information for all figures if necessary.

Figure 6: All panels are almost the same and I don't understand the whole paragraph of discussing Figure 6 in section 3.2. For example, when you are trying to explain the nonlinear relationship, you should try to use tables rather than explaining the figure. In addition, in L172-174, you stated "When the AESF increases from 0.125 to 1.0, near-surface [PM2.5] increase by 6.6 times. However, when the AESF increases from 1.0 to 8.0, the enhancement of [PM2.5] is 9.4 times". This is more likely to describe a proportional relationship rather than linear relationship, even though the proportional relationship is a special linear relationship. Also, your assumed relationship is supposed to be explained earlier, i.e. why do you believe near-surface [PM2.5] should be 8 times larger when AESF is 8 times larger. How is that AESF working in the model?

Figure 12 and Figure 13: Given that you want to compare the area and the amount of precipitation, you can try to sum up the total amount of precipitation in the selected domain (total: x mm, south with some definition about the region: y mm) and count the grids number for each color (yellow: x, red: y, …). I tried to compare these two figures back and forth, and found it difficult to get the same results as you mentioned in the manuscript.

[Specific Comments]

L31-32: The sentence is ambiguous. Typically, we don't say cloud intensity, so you shouldn't say "their", but you are supposed to specifically point out which variable you are mentioning. Also, if "intensity" is about precipitation, why do you mention "precipitation amount" in the end?

L106-107: What is "designed with a strech"? Are you sure this is the term we describe the uneven vertical grid spacing? Do you mean that the vertical resolutions are different, i.e. 30 m near surface and 400 m above 2.5 km? You can be more specific by mentioning "vertical resolution" to make it clear if so.

L110: Are you sure this is a 60-h period? Do you have some spin-up time not included, otherwise you made a mistake here?

L121: I am confused here about "benchmark simulation". Is it just the CTRL result? If so, you may just say "control simulation results" or other name about "control" rather than involving a new word "benchmark".

L140: You may mention this is local time if so. Model simulations usually use UTC time, as well as in your manuscript.

L145-148: The maximum CAPE of 5045 J/kg is indeed really high. However, it is not safe to say the CAPE decreased significantly after the rainfall. First, you need to zoom in the CAPE values from the title to the description of the figure, otherwise they are very hard to be found. Second, the CAPE observations occur 2 times per day, whose time resolution much longer than a convective precipitation event. Third, all three CAPE observations are high (4416, 5045 and 2486 J/kg, and please make them larger). You cannot argue that CAPE decreases a lot after the rainfall. How do you know that the decreasing doesn't result from the sunset (UTC 1200 is local time 2000 sunset time, also add the local time information as I state in the general comments)? You can't simply compare 4416 with 2486, unless you have historical observation data showing 4416 is higher than average but 2486 isn't. In short, you can say UTC 1200 CAPE observational value is very high, but you can't specifically guess its trend or changing reason. By the way, Figure S1 and Figure S2 can be added in the main text.

L155: You may add this sentence "IOA describes the relative difference between the model and observation, ranging from 0 to 1, with 1 indicating perfect agreement" from the supplement to the main text, so that the readers can understand the IOA values of 0.88 and 0.96 are really high.

Figure 3: Panel (c) and (d) don't have that high IOAs and do you have a specific reason about them? Maybe you can mention this in the main text.

L165: How do you know starting from 0400 UTC "clouds commence to form and develop"? Do you have observations or your model simulations to support this statement?

L193-194: I can't see the relationship between increased stability and ARI effects on water vapor mixing ratio. I believe that it is more related with Figure 7a: With lower atmosphere gets cooler and upper atmosphere gets warmer, it is likely to be more stable. If so, you could move this sentence forward a bit.

L195: Why does the "warm bubble" effect appear? Is it related with upper atmosphere getting heated, like Figure 7a from 1000 m to 2500 m? If so, please related Figure 7a with Figure 7c.

Figure 8: You mentioned a lot of thresholds of AESF, such as 0.33, 1.6, …, and I hope that you can label these values on the x-axis (same suggestion for all similar figures) so that the readers can find them out quickly. Also, your statements about panel (c) and (d) are so complicated that I can't understand why there are so many intervals. From my point of view, I can only see that the red dots are significantly lower than the blue dots when AESF is more than 1.6. The differences between red and blue dots are too small to be trusted when AESF is less than 1.6, unless you can prove your results are robust.

Figure 9: Similar with Figure 8. There are too many intervals and I can only believe significant differences appear when AESF exceeding 1.6 from panel (a) and (c). Otherwise, when AESF is smaller than 1.6, the differences of updraft velocity only range from -4 % to 4 %, and I am more likely to believe it is caused by

model itself. I am wondering if you can really prove your results are robust when AESF is small.

L245-247: Yes, I can see the strong correlation between CWP and updraft, which makes sense. But I don't understand why you say "However, the variation direction of the two variables is not always consistent." The correlation coefficient of 0.87 is indeed very high, which can explain why the red dots in Figure 9 (a) and (c) have almost the same tendency. The following discussion in this paragraph doesn't disprove this consistency, so you don't need to start with "however".

L247-255: You try to explain the negative correlation between CDNC and updrafts, but it seems to be too complicated and I think you it should be the relationship between CDNC and AESF (or aerosol itself) rather than updrafts. That's why I am confused. I suggest you can start with your Figure 10 and talk about the reverse relationship between aerosol loading (no matter ARI included or not) and Reff. Then you need to prove the relationship between Reff and the conversion efficiency or CDNC.

L268-269: You haven't found that higher aerosol loading resulting in an increasing precipitation and I don't know whether it is related with the scale of the convection. Like you said earlier, the GZB during the event is located near the bottom of a trough at 850 hPa and the center of 200 hPa. Also, according to Figure 5, the precipitation area was too large to be a single convective cell. I am wondering if the precipitation itself is more likely to be determined by dynamics rather than aerosol itself. If so, the convective precipitation amount won't change a lot unless the aerosol loading is too high.

L287: How do you define "heavy and torrential rainfall" or do you have any thresholds? It would be better if you can state specific color(s) in the figures as heavy rainfall, like yellow and red. Also, Figure 12 and Figure 13 should be combined, so that the readers can compare them directly.

L290-294: Even though ARIs can alter the precipitation amount and area, how can you determine the change is caused by the lifting or stabilizing effects? Are they considered as the only effects on convective precipitations and could you please justify this?

L306: Do you want to compare Figure 14 (b) and (d)?

L325-326: How do you consider the effect of ACIs in your simulation? Are you referring to Figure 8? If so, you can be more specific, as ACIs are much more complicated, including the cloud lifetime, the ratio between ice and liquid, etc. You may also revise the abstract by specifying the ACIs.

[Technical Corrections]

General: It would be better if you can put the figures in the main text, so that the readers don't need to turn the pages back and forth to find out which figure you are talking about.

L99: What is SI of Supplement Information?

L110: The time information is a little bit confused. Could you please revise it to "from UTC 1200 of July 21 to UTC 0000 of

July 25, 2016" if this is not convention? Actually, I was a bit confused when I saw UTC 25 or so. Same for L118. You may keep it if you are sure this is conventional writing style.

Figure 2: There is a strange symbol in the text of Figure 2. Also see Figure 5.

Figure 7: Larger font size is preferred here.

L187: …, the deduction becomes more pronounced with increasing AESF, but is not sensitive to height.

---

## Author Comment (AC1)

**Reply to Anonymous Referee #1**

We thank the reviewer for the careful reading of the manuscript and helpful comments. We have revised the manuscript following the suggestion, as described below.

1. Time Zone: I was confused by time zones throughout the manuscript. When you are talking the results of model simulations, I know they should be based on UTC by default. But when you are talking about observations, it is little tricky because I don't know whether they are using the local time. Also, you may add the local time next to UTC time, otherwise it will be a confusion. For example, L160 mentioned morning time as well as from 0000 to 0400 UTC, the readers have to convert UTC time to local time to understand you are talking about from 0800 to 1200 local time. Also, you may add all local time information for all figures if necessary.

**Response: We have added the local time next to UTC time throughout the manuscript:**

L132-133: "*The WRF-Chem is first integrated for an 84-h period from 1200 UTC (2000 LT) of July 21 to 0000 UTC (0800 LT) of July 25, 2016 for D01, with a 30-h spin-up time.*"

L146-147: "*The WRF-Chem model is then integrated for a 24-h period from 0000 UTC (0800 LT) of July 24 to 0000 UTC (0800 LT) of July 25, 2016 for D02.*"

L181-185: "*The high convective available potential energy (CAPE) of 5045 J kg$^{-1}$ is observed at 1200 UTC (2000 LT) 24 July (Figure 3b), which is right before the heavy rainfall peak (1400 UTC / 2200 LT) in Xi'an. The CAPE decreases to 2486 J kg$^{-1}$ at 0000 UTC (0800 LT) 25 July (Figure 3c), which is plausibly attributed to post-precipitation stabilization through latent heat release and nocturnal surface cooling following sunset.*"

L199-200: "*Figure 4 presents the spatial distribution of simulated and observed concentrations of PM$_{2.5}$, O$_3$, NO$_2$, and SO$_2$ at 0000 UTC (0800 LT) on 24 July 2016.*"

L208-210: "*NO$_2$ over-predictions appear primarily from 1200 UTC to 1800 UTC (2000 – 0200 LT), which results from the uncertainty of nighttime traffic emission inventory and the possibly under-predicted plenary boundary layer.*"

L225-226: "*For example, the enhancement of rain rate from 1000 to 1400 UTC (from 1800 to 2200 LT) is reproduced, and the rapid falloff from 1400 to 1800 UTC (from 2200 LT on 24 July to 0200 LT on 25 July) is simulated.*"

L247-250: "*We first examine ARI effects on the profile of temperature and water vapor in the morning (from 0000 to 0400 UTC / from 0800 to 1200 LT), since after 0400 UTC (1200 LT) near-surface [PM$_{2.5}$] start decreasing (Figure 5a) and the clouds commence to form and develop, with occurrence of sporadic precipitation in the GZB (Figure 6a).*"

L274: "*Figure 9a provides the ARI effect on the average temperature profile from 0000 to 0400 UTC (from 0800 to 1200 LT) over the GZB by comparing the F_BASE and F_ARI0 under different aerosol conditions.*"

L304-305: "*We then investigate the effect of ACIs and ARIs on cloud properties and precipitation during the main precipitation period from 0800 to 1800 UTC (from 1600 LT on 24 July to 0200 LT on 25 July).*"

**We have added local time information for all related figures:**

[Figure]

Figure 5: Comparison of observed (black dots) and simulated (solid red lines) diurnal profile of near-surface hourly mass concentrations of (a) PM2.5, (b) O₃, (c) NO₂, and (d) SO₂ averaged at monitoring sites in the GZB on 24 July 2016.

[Figure]

Figure 6: Comparison of observed (black dots) and simulated (solid red lines) diurnal profile of hourly rain rate averaged at monitoring sites in the (a) GZB and (b) GZBs on 24 July 2016.

2. Figure 6: All panels are almost the same and I don't understand the whole paragraph of discussing Figure 6 in section 3.2. For example, when you are trying to explain the nonlinear relationship, you should try to use tables rather than explaining the figure. In addition, in L172-174, you stated "When the AESF increases from 0.125 to 1.0, near-surface [PM2.5] increase by 6.6 times. However, when the AESF increases from 1.0 to 8.0, the enhancement of [PM2.5] is 9.4 times". This is more likely to describe a proportional relationship rather than linear relationship, even though the proportional relationship is a special linear relationship. Also, your assumed relationship is supposed to be explained earlier, i.e. why do you believe near-surface [PM2.5] should be 8 times larger when AESF is 8 times larger. How is that AESF working in the model?

**Response:** We have added a table (Table S2) and revised the sentence in L254-255: "*Near-surface PM$_{2.5}$ concentrations monotonically increase with increasing anthropogenic emissions as expected, showing a proportional relationship. With the increasing AESF or anthropogenic emissions, the increasing rate of near-surface [PM$_{2.5}$] is enhanced (Table S2). For example, when the AESF increases from 0.125 to 1.0, near-surface [PM$_{2.5}$] increase by 6.6 times. However, when the AESF increases from 1.0 to 8.0, the enhancement of [PM$_{2.5}$] is 9.4 times, which is mainly caused by the ARI effect which suppresses development of the PBL to increase near-surface air pollutants level and the enhanced formation of secondary aerosols (Wu et al., 2019).*"

Table S2 Variations of near-surface PM$_{2.5}$ and BC concentrations, AOD, and AAOD in the morning in the GZB with the AESF.

| AESF | | PM$_{2.5}$ | | BC | | AOD | | AAOD | |
|---|---|---|---|---|---|---|---|---|---|
| Index | Value | Concentration ($\mu$g m$^{-3}$) | Rate[1] | Concentration ($\mu$g m$^{-3}$) | Rate | Value | Rate | Value | Rate |
| 1 | 0.125 | 4.8 | | 0.21 | | 0.064 | | 0.0048 | |
| 2 | 0.139 | 5.2 | 28.13 | 0.23 | 1.65 | 0.070 | 0.432 | 0.0053 | 0.0391 |
| 3 | 0.154 | 5.7 | 28.25 | 0.25 | 1.65 | 0.077 | 0.470 | 0.0059 | 0.0392 |
| 4 | 0.171 | 6.1 | 28.37 | 0.28 | 1.65 | 0.085 | 0.455 | 0.0066 | 0.0393 |
| 5 | 0.189 | 6.7 | 28.49 | 0.31 | 1.66 | 0.093 | 0.432 | 0.0073 | 0.0395 |
| 6 | 0.210 | 7.3 | 28.61 | 0.35 | 1.66 | 0.101 | 0.426 | 0.0081 | 0.0395 |
| 7 | 0.233 | 7.9 | 28.86 | 0.38 | 1.66 | 0.111 | 0.432 | 0.0091 | 0.0395 |
| 8 | 0.259 | 8.7 | 28.97 | 0.43 | 1.66 | 0.122 | 0.432 | 0.0101 | 0.0395 |
| 9 | 0.287 | 9.5 | 29.13 | 0.47 | 1.66 | 0.134 | 0.423 | 0.0112 | 0.0395 |
| 10 | 0.319 | 10.4 | 29.44 | 0.53 | 1.67 | 0.148 | 0.426 | 0.0124 | 0.0395 |
| 11 | 0.354 | 11.5 | 29.58 | 0.59 | 1.67 | 0.163 | 0.428 | 0.0138 | 0.0395 |
| 12 | 0.392 | 12.6 | 29.95 | 0.65 | 1.67 | 0.179 | 0.422 | 0.0153 | 0.0396 |
| 13 | 0.435 | 13.9 | 30.09 | 0.72 | 1.67 | 0.197 | 0.424 | 0.0170 | 0.0396 |
| 14 | 0.483 | 15.4 | 30.38 | 0.80 | 1.68 | 0.218 | 0.426 | 0.0189 | 0.0395 |
| 15 | 0.536 | 17.0 | 30.74 | 0.89 | 1.68 | 0.240 | 0.427 | 0.0210 | 0.0396 |
| 16 | 0.595 | 18.8 | 31.03 | 0.99 | 1.69 | 0.265 | 0.426 | 0.0234 | 0.0396 |
| 17 | 0.660 | 20.9 | 31.35 | 1.10 | 1.69 | 0.293 | 0.428 | 0.0259 | 0.0396 |
| 18 | 0.732 | 23.1 | 31.62 | 1.22 | 1.70 | 0.324 | 0.424 | 0.0288 | 0.0397 |
| 19 | 0.812 | 25.7 | 32.02 | 1.36 | 1.70 | 0.358 | 0.426 | 0.0320 | 0.0396 |
| 20 | 0.901 | 28.6 | 32.34 | 1.51 | 1.71 | 0.395 | 0.421 | 0.0355 | 0.0397 |
| 21 | 1.000 | 31.8 | 32.69 | 1.68 | 1.72 | 0.438 | 0.427 | 0.0394 | 0.0397 |
| 22 | 1.110 | 35.4 | 33.09 | 1.87 | 1.72 | 0.484 | 0.422 | 0.0438 | 0.0398 |
| 23 | 1.231 | 39.5 | 33.36 | 2.08 | 1.73 | 0.535 | 0.422 | 0.0486 | 0.0398 |
| 24 | 1.366 | 44.0 | 33.71 | 2.31 | 1.74 | 0.592 | 0.420 | 0.0540 | 0.0399 |
| 25 | 1.516 | 49.1 | 34.08 | 2.58 | 1.75 | 0.655 | 0.423 | 0.0600 | 0.0399 |
| 26 | 1.682 | 54.9 | 34.42 | 2.87 | 1.76 | 0.726 | 0.429 | 0.0666 | 0.0399 |

| | | | | | | | | | |
|---|---|---|---|---|---|---|---|---|---|
| 27 | 1.866 | 61.3 | 34.82 | 3.20 | 1.77 | 0.804 | 0.420 | 0.0740 | 0.0400 |
| 28 | 2.071 | 68.5 | 35.21 | 3.56 | 1.79 | 0.889 | 0.415 | 0.0822 | 0.0401 |
| 29 | 2.297 | 76.6 | 35.63 | 3.97 | 1.81 | 0.983 | 0.415 | 0.0913 | 0.0401 |
| 30 | 2.549 | 85.6 | 35.96 | 4.43 | 1.83 | 1.089 | 0.423 | 0.1014 | 0.0402 |
| 31 | 2.828 | 95.7 | 36.27 | 4.95 | 1.85 | 1.205 | 0.414 | 0.1126 | 0.0402 |
| 32 | 3.138 | 107.1 | 36.63 | 5.53 | 1.87 | 1.332 | 0.409 | 0.1251 | 0.0403 |
| 33 | 3.482 | 119.9 | 37.17 | 6.18 | 1.90 | 1.474 | 0.415 | 0.1390 | 0.0405 |
| 34 | 3.864 | 134.1 | 37.42 | 6.91 | 1.92 | 1.633 | 0.416 | 0.1545 | 0.0404 |
| 35 | 4.287 | 150.2 | 38.03 | 7.74 | 1.96 | 1.811 | 0.421 | 0.1716 | 0.0405 |
| 36 | 4.757 | 168.3 | 38.33 | 8.67 | 1.98 | 2.006 | 0.413 | 0.1907 | 0.0406 |
| 37 | 5.278 | 188.4 | 38.71 | 9.72 | 2.01 | 2.217 | 0.406 | 0.2119 | 0.0407 |
| 38 | 5.856 | 211.1 | 39.26 | 10.91 | 2.06 | 2.450 | 0.403 | 0.2355 | 0.0408 |
| 39 | 6.498 | 236.8 | 40.06 | 12.27 | 2.12 | 2.715 | 0.412 | 0.2617 | 0.0409 |
| 40 | 7.210 | 266.0 | 40.95 | 13.82 | 2.18 | 3.008 | 0.412 | 0.2909 | 0.0410 |
| 41 | 8.000 | 299.2 | 42.05 | 15.61 | 2.26 | 3.328 | 0.404 | 0.3234 | 0.0411 |

[1]Rate is defined as the variation rate of a variable with the AESF, which is expressed as: $rate(j,i) = \frac{V_i(j)-V_i(j-1)}{AESF(j)-AESF(j-1)}$, where $j$ denotes the sensitivity simulation index and $j$ = 1, 2, …, 41; $i$ = 1, 2, …, 4, and $V_i$ represents near-surface PM$_{2.5}$ and BC concentrations, AOD, and AAOD in the morning in the GZB, respectively.

3. Figure 12 and Figure 13: Given that you want to compare the area and the amount of precipitation, you can try to sum up the total amount of precipitation in the selected domain (total: x mm, south with some definition about the region: y mm) and count the grids number for each color (yellow: x, red: y, …). I tried to compare these two figures back and forth, and found it difficult to get the same results as you mentioned in the manuscript.

**Response:** We have combined Figure 12 and 13 (Figure 14 in the revised version) to facilitate comparison. We have also changed the sentence in L458-470 "*Comparing Figure 14d0 and Figure 14d1, the ARI effect also increases considerably the area with heavy rainfall, but the precipitation in the south (the area surrounded by the white rectangle) of the GZB is decreased. The average daily precipitation is 16.7 and 15.8 mm d$^{-1}$ in the GZB, and 9.6 and 13.9 mm d$^{-1}$ in the south area with and without the ARI effect, respectively. The grid number with occurrence of heavy and torrential rainfall is 878 and 774 in the GZB with the ARI effect, and 802 and 721 without the ARI effect, respectively. In the south area, the number is 824 and 170 with the ARI effect, and 1125 and 381 without the ARI effect, respectively. When the lifting effect outweighs the stabilizing effect due to ARIs in the GZB, the increased vertical wind speed (as evidenced by enhanced vertical velocities in Figure 11b) induces the horizontal convergence above the PBL, causing the transport of water vapor from its surroundings, which enhances precipitation in the GZB but reduces precipitation in its surroundings. The additional contributing mechanisms also exist, including but not limited to the effect of ARIs-induced aerosol spatial heterogeneity on thermodynamic-dynamic fields, cloud and precipitation processes. These complex interactions warrant further investigation through targeted sensitivity experiments.*"

[Specific Comments]
4. L31-32: The sentence is ambiguous. Typically, we don't say cloud intensity, so you shouldn't say "their", but you are supposed to specifically point out which variable you are mentioning. Also, if "intensity" is about precipitation, why do you mention "precipitation amount" in the end?

**Response:** We have revised the sentence in L33-36 as "*Atmospheric aerosols influence both cloud*

*processes (e.g., initiation time, lifetime, and spatial extent) and precipitation characteristics (including duration, frequency, and cumulative amount), with these coupled interactions remaining the dominant source of uncertainty in quantifying climate forcing agents and refining future scenarios (IPCC, 2013).*"

5. L106-107: What is "designed with a strech"? Are you sure this is the term we describe the uneven vertical grid spacing? Do you mean that the vertical resolutions are different, i.e. 30 m near surface and 400 m above 2.5 km? You can be more specific by mentioning "vertical resolution" to make it clear if so.

**Response:** We have revised the statement to make it clear in L122-124 as "*The model employs vertically staggered grids with enhanced resolution near the ground surface (30 m vertical spacing), increasing to progressively coarser resolution at higher elevations (reaching 400 m grid spacing above 2.5 km).*"

6. L110: Are you sure this is a 60-h period? Do you have some spin up time not included, otherwise you made a mistake here?

**Response:** We have corrected the sentence in L132-133 as "*The WRF-Chem is first integrated for an 84-h period from 1200 UTC (2000 LT) of July 21 to 0000 UTC (0800 LT) of July 25, 2016 for D01, with a 30-h spin-up time.*"

7. L121: I am confused here about "benchmark simulation". Is it just the CTRL result? If so, you may just say "control simulation results" or other name about "control" rather than involving a new word "benchmark".

**Response:** We have revised it in L148 as "*Both ARI and ACI effects are considered in the control simulation, in...*"

8. L140: You may mention this is local time if so. Model simulations usually use UTC time, as well as in your manuscript.

**Response:** We have revied the sentence in L176 as "*The selected heavy rainfall event occurred on the local time of July 24-25, 2016 in the GZB.*"

9. L145-148: The maximum CAPE of 5045 J/kg is indeed really high. However, it is not safe to say the CAPE decreased significantly after the rainfall. First, you need to zoom in the CAPE values from the title to the description of the figure, otherwise they are very hard to be found. Second, the CAPE observations occur 2 times per day, whose time resolution much longer than a convective precipitation event. Third, all three CAPE observations are high (4416, 5045 and 2486 J/kg, and please make them larger). You cannot argue that CAPE decreases a lot after the rainfall. How do you know that the decreasing doesn't result from the sunset (UTC 1200 is local time 2000 sunset time, also add the local time information as I state in the general comments)? You can't simply compare 4416 with 2486, unless you have historical observation data showing 4416 is higher than average but 2486 isn't. In short, you can say UTC 1200 CAPE observational value is very high, but you can't specifically guess its trend or changing reason. By the way, Figure S1 and Figure S2 can be added in the main text.

**Response:** We have revised the figure and related discussions in L181-185: "*The high convective available potential energy (CAPE) of 5045 J kg$^{-1}$ is observed at 1200 UTC (2000 LT) 24 July (Figure 3b), which is right before the heavy rainfall peak (1400 UTC / 2200 LT) in Xi'an. The CAPE decreases to 2486 J kg$^{-1}$ at 0000 UTC (0800 LT) 25 July (Figure 3c), which is plausibly attributed to post-precipitation stabilization through latent heat release and nocturnal surface cooling following sunset.*"

[Figure]

**Figure 3: Atmospheric sounding over the GZB (108.97°E, 34.43°N) at (a) 0000 UTC and (b) 1200 UTC (2000 LT) on 24, and (c) 0000 UTC (0080 LT) on 25 July, 2016. The black line denotes the temperature, and the blue line represents the dew point temperature.**

10. L155: You may add this sentence "IOA describes the relative difference between the model and observation, ranging from 0 to 1, with 1 indicating perfect agreement" from the supplement to the main text, so that the readers can understand the IOA values of 0.88 and 0.96 are really high.

**Response:** We have added the sentence in L162-163 of the main text as "*IOA describes the relative difference between the model and observation, ranging from 0 to 1, with 1 indicating perfect agreement.*"

11. Figure 3: Panel (c) and (d) don't have that high IOAs and do you have a specific reason about them? Maybe you can mention this in the main text.

**Response:** We have added discussions related to the simulated bias of the panel (c) and (d) in L207-211 of the text: "*The model overestimates $NO_2$ by 0.76 µg m$^{-3}$ and underestimates $SO_2$ by 0.23 µg m$^{-3}$, with the IOA of 0.68 and 0.51, respectively (Figure 5c, d). $NO_2$ over-predictions appear primarily from 1200 UTC to 1800 UTC (2000 – 0200 LT), which results from the uncertainty of nighttime traffic emission inventory and the possibly under-predicted plenary boundary layer. $SO_2$ bias mainly stem from point sources, which is highly sensitive to uncertainties in simulated wind fields.*"

12. L165: How do you know starting from 0400 UTC "clouds commence to form and develop"? Do you have observations or your model simulations to support this statement?

**Response:** We have clarified in L247-250: "*We first examine ARI effects on the profile of temperature and water vapor in the morning (from 0000 to 0400 UTC / from 0800 to 1200 LT), since after 0400 UTC (1200 LT) near-surface [PM$_{2.5}$] start decreasing (Figure 5a) and the clouds commence to form and*"

*develop, with occurrence of sporadic precipitation in the GZB (Figure 6a).***"**

13. L193-194: I can't see the relationship between increased stability and ARI effects on water vapor mixing ratio. I believe that it is more related with Figure 7a: With lower atmosphere gets cooler and upper atmosphere gets warmer, it is likely to be more stable. If so, you could move this sentence forward a bit.

**Response:** We have moved the sentence forward in L281-282: "*Meanwhile, the perturbation of temperature profile caused by ARIs also suppresses development of the PBL, which does not facilitate dispersion of air pollutants and water vapor in the PBL. Therefore, the ARI effect increases the atmospheric stability, which tends to inhibit cloud formation and development. The ARI effect increases the mass mixing ratio of water vapor in the atmosphere below around 500 m and decreases it in the atmosphere from about 500 m to 1700 m (Figure 9b).*"

14. L195: Why does the "warm bubble" effect appear? Is it related with upper atmosphere getting heated, like Figure 7a from 1000 m to 2500 m? If so, please related Figure 7a with Figure 7c.

**Response:** We have revised the sentence in L291-292 as: "*The temperature enhancement caused by absorbing aerosols above the PBL cause a "warm bubble" effect (Figure 9a) (Wu et al., 2025), which could induce updrafts to promote convection. As shown in Figure 9c, the heating effect of ARIs generates a secondary upward movement in the atmosphere above around 300m.*"

*Wu, J., Bei, N., Wang, Y., Su, X., Zhang, N., Wang, L., Hu, B., Wang, Q., Jiang, Q., Zhang, C., Liu, Y., Wang, R., Li, X., Lu, Y., Liu, Z., Cao, J., Tie, X., Li, G., and Seinfeld, J.: Aerosol light absorption alleviates particulate pollution during wintertime haze events, Proc. Natl. Acad. Sci. U.S.A., 122, e2402281121, doi: 10.1073/pnas.2402281121, 2025.*

15. Figure 8: You mentioned a lot of thresholds of AESF, such as 0.33, 1.6, …, and I hope that you can label these values on the xaxis (same suggestion for all similar figures) so that the readers can find them out quickly. Also, your statements about panel (c) and (d) are so complicated that I can't understand why there are so many intervals. From my point of view, I can only see that the red dots are significantly lower than the blue dots when AESF is more than 1.6. The differences between red and blue dots are too small to be trusted when AESF is less than 1.6, unless you can prove your results are robust.

**Response:** We have labelled the threshold values in Figures 8-11, and 14 (Figures 10-13, and 15 in the revised version) and revised the sentence in L318-319: "*In addition, only when the AESF is in the range between 0.27 and 0.70, the ARI effect increases the CWP (Figure 8d).*" to "*In addition, when the AESF is less than 1.6, the ARI effect increases or decreases the CWP by up to 5% (Figure 10d).*", and in L329-330: "*When the AESF is in the range between 0.33 and 0.70, the ARI effect simultaneously increases the CDNC and CWP. With the AESF exceeding 0.70, the ARI effect increases the CDNC but decreases CWP.*" To "*When the AESF is more than 1.6, the ARI effect considerably increases the CDNC but decreases CWP.*"

[Figure]

Figure 10: (a) p-mean of CDNC, (b) variation of p-mean of CDNC due to ARIs, (c) average CWP, and (d) variation of CWP due to ARIs over the GZB from 0800 to 1800 UTC (from 1600 LT on 24 July to 0200 LT on 25 July), as a function of the scale factor of anthropogenic emissions.

[Figure]

Figure 11: Average (a) updraft, (b) variation of updraft due to ARIs, (c) IWP, and (d) variation of IWP due to ARIs over the GZB from 0800 to 1800 UTC (from 1600 LT on 24 July to 0200 LT on 25 July), as a function of the scale factor of anthropogenic emissions.

[Figure]

**Figure 12: Average (a) Reff of cloud droplets and (b) variation of Reff due to ARIs over the GZB from 0800 to 1800 UTC (from 1600 LT on 24 July to 0200 LT on 25 July), as a function of the scale factor of anthropogenic emissions.**

[Figure]

**Figure 13: Average (a) accumulative precipitation and (b) variation of precipitation due to ARIs in the GZB on 24 July 2016, as a function of the scale factor of anthropogenic emissions.**

[Figure]

**Figure 15: Average (a) daily precipitation, and (b) variation of daily precipitation, (c) updraft and (d) variation of updraft due to ARIs from 0800 to 1800 UTC (from 1600 LT on 24 July to 0200 LT on 25 July) over the whole domain, as a function of the scale factor of anthropogenic emissions.**

16. Figure 9: Similar with Figure 8. There are too many intervals and I can only believe significant

differences appear when AESF exceeding 1.6 from panel (a) and (c). Otherwise, when AESF is smaller than 1.6, the differences of updraft velocity only range from -4 % to 4 %, and I am more likely to believe it is caused by model itself. I am wondering if you can really prove your results are robust when AESF is small.

**Response:** We have revised Figure 9 (Figure 11 in the revised version) and associated discussions as suggested. In L350-351, we have changed the sentence "*The ARI effect enhances the updraft in the GZB with the AESF of 0.125 and in the range between 0.4 and 1.6, and weakens it under other AESF conditions (Figure 9b).*" to "*The ARI effect modulates the updraft in the GZB with the AESF less than 1.6, and the updraft variation due to the ARI effect is in the range between -4% to 4% (Figure 11b).*". And we also changed the sentence in L372-373: "*When the AESF is more than 2.0, the ARI effect decreases the updraft by more than 6% and the updraft decrease generally becomes increasingly significant with increasing AESF.*" to "*When the AESF is more than 1.6, the ARI effect decreases the updraft consistently and the updraft decrease generally becomes increasingly significant with increasing AESF.*"

[Figure]

**Figure 11: Average (a) updraft, (b) variation of updraft due to ARIs, (c) IWP, and (d) variation of IWP due to ARIs over the GZB from 0800 to 1800 UTC (from 1600 LT on 24 July to 0200 LT on 25 July), as a function of the scale factor of anthropogenic emissions.**

17. L245-247: Yes, I can see the strong correlation between CWP and updraft, which makes sense. But I don't understand why you say "However, the variation direction of the two variables is not always consistent." The correlation coefficient of 0.87 is indeed very high, which can explain why the red dots in Figure 9 (a) and (c) have almost the same tendency. The following discussion in this paragraph doesn't disprove this consistency, so you don't need to start with "however".

**Response:** We have removed the sentence in L383: "*However, the variation direction of the two variables is not always consistent.*".

18. L247-255: You try to explain the negative correlation between CDNC and updrafts, but it seems to be too complicated and I think you it should be the relationship between CDNC and AESF (or aerosol itself) rather than updrafts. That's why I am confused. I suggest you can start with your Figure 10 and talk about the reverse relationship between aerosol loading (no matter ARI included or not) and Reff. Then you need to prove the relationship between Reff and the conversion efficiency or CDNC.

**Response:** We sincerely appreciate the reviewer's insightful comments. We acknowledge that the original phrasing in L247-255 may have caused ambiguity regarding the causal relationships. To clarify, our analysis specifically focuses on the anti-correlation between ARI-induced CDNC changes and ARI-induced updraft modifications, rather than the direct relationship between absolute CDNC and updraft velocities. We have revised the statement in L383-384 as "*The ARI-induced perturbations reveal a negative correlation (r = -0.86) between aerosol-mediated CDNC variations and updraft variations.*". We are sorry for the typo error in L392 of the manuscript: "*The variation of CDNC due to ARIs is highly correlated with that of $R_{eff}$, with a correlation coefficient of about 0.98.*", which makes the discussions confusing. We have corrected it to "*The variation of CDNC due to ARIs is highly correlated with that of $R_{eff}$, with a correlation coefficient of about -0.98.*"

19. L268-269: You haven't found that higher aerosol loading resulting in an increasing precipitation and I don't know whether it is related with the scale of the convection. Like you said earlier, the GZB during the event is located near the bottom of a trough at 850 hPa and the center of 200 hPa. Also, according to Figure 5, the precipitation area was too large to be a single convective cell. I am wondering if the precipitation itself is more likely to be determined by dynamics rather than aerosol itself. If so, the convective precipitation amount won't change a lot unless the aerosol loading is too high.

**Response:** We fully agree with the reviewer's perspective that large-scale dynamical forcing likely dominated precipitation production in this case, especially given the synoptic setup (850 hPa trough and 200 hPa divergence in Figure 2) that provided persistent upward motion. This aligns with our discussions that aerosol effects on precipitation exhibit threshold-dependent behavior in such dynamically dominated systems: when AESF > 1.0: High loading aerosols (more CCN) with relatively not abundant water vapor will lead competitions between convective clouds for water vapor, which reduces droplet sizes, inhibits the glacier process, weakens convections, causing decrease of precipitation.

We have revised the discussions in L423-426 and L431-432 as follows, "*We do not observe significant increasing trend of precipitation with increasing AESF in both F_BASE and F_ARI0. This discrepancy likely stems from the unique dynamical context of our case: the MCS developed under strong synoptic forcing (850 hPa trough and 200 hPa divergence in Figure 2), where large-scale moisture convergence dominated precipitation production, effectively masking aerosol microphysical effects at moderate loading (AESF <1.0). In addition, when the AESF is more than 1.0, the decreasing trend of precipitation with increasing AESF becomes significant. Elevated aerosols increase CDNC and cloud water content and reduces droplet size to inhibit autoconversion, enhancing glacier processes to invigorate convection and further precipitation. However, when the droplet size is decreased to a threshold due to increased aerosols, the glacier process is inhibited and convection commences to be weakened, causing decrease of precipitation. This threshold behavior (AESF >1.0) emerges only when aerosol-induced microphysical suppression overwhelms the dynamical moisture supply capacity. Although our simulations show that*

*CDNC and CWP increase monotonically with increasing AESF, the total ice-phase hydrometeors and updrafts do not have significant increasing trend with the AESF less than 1.0. When the AESF is more than 1.0, those two variables exhibit decreasing trend with the AESF. As discussed above, the main reason is competition between convective clouds to available water vapor in the MCS.*"

20. L287: How do you define "heavy and torrential rainfall" or do you have any thresholds? It would be better if you can state specific color(s) in the figures as heavy rainfall, like yellow and red. Also, Figure 12 and Figure 13 should be combined, so that the readers can compare them directly.

**Response:** We have added labels, revised and combined the figures (shown by Figure 14 in the revised version), and added the definition of precipitation intensity in L449-453: "*Daily precipitation intensity is classified following the China Meteorological Administration standard into five categories: light (0.1-9.9 mm $d^{-1}$), moderate (10-24.9 mm $d^{-1}$), heavy (25-49.9 mm $d^{-1}$), torrential (50-99.9 mm $d^{-1}$), and downpour ($\geq$100 mm $d^{-1}$) rainfall (Ma et al., 2015), which are visually distinguished in the figure through different colors.*"

*Ma, S., Zhou, T., Dai, A., and Han, Z.: Observed changes in the distributions of daily precipitation frequency and amount over China from 1960 to 2013, J. Climate, 28, 6960-6978, doi: 10.1175/JCLI-D-15-0011.1, 2015.*

[Figure]

**Figure 14: Accumulative precipitation distribution on 24 July 2016 for various scale factor of anthropogenic emissions when ARIs are not considered (*0), and (*1) are considered.**

[Figure]

**Figure 14: continued**

[Figure]

**Figure 14: continued**

21. L290-294: Even though ARIs can alter the precipitation amount and area, how can you determine the change is caused by the lifting or stabilizing effects? Are they considered as the only effects on convective precipitations and could you please justify this?

**Response:** We have revised the added discussions in L464-470: "*When the lifting effect outweighs the stabilizing effect due to ARIs in the GZB, the increased vertical wind speed (as evidenced by enhanced vertical velocities in Figure 11b) induces the horizontal convergence above the PBL, causing the transport of water vapor from its surroundings, which enhances precipitation in the GZB but reduces precipitation in its surroundings. The additional contributing mechanisms also exist, including but not limited to the effect of ARIs-induced aerosol spatial heterogeneity on thermodynamic-dynamic fields,*

*cloud and precipitation processes. These complex interactions warrant further investigation through targeted sensitivity experiments.*"

22. L306: Do you want to compare Figure 14 (b) and (d)?

**Response:** We have corrected the statement in L507 as "*The variation of p-mean of the updraft due to ARIs in the domain is highly correlated with that of the precipitation, with the correlation coefficient of 0.96 (Figures 15b, d).*"

23. L325-326: How do you consider the effect of ACIs in your simulation? Are you referring to Figure 8? If so, you can be more specific, as ACIs are much more complicated, including the cloud lifetime, the ratio between ice and liquid, etc. You may also revise the abstract by specifying the ACIs.

**Response:** To address the effect of ACIs, we have conducted the F_ARI0 series of experiments, which excluded ARIs. This approach allowed us to investigate the impact of ACIs on precipitation. We have refined the abstract by specifying the role of ACIs in L22 of the abstract as "*When the ARI effect is not considered, the daily precipitation does not show an increasing trend with increasing aerosols in the GZB. This primarily reflects the effects of ACIs due to competition among convective clouds to available water vapor in development of the MCS.*"

**Technical Corrections**

24. General: It would be better if you can put the figures in the main text, so that the readers don't need to turn the pages back and forth to find out which figure you are talking about.

**Response:** We have put the figures in the main test as suggested.

25. L99: What is SI of Supplement Information?

**Response:** It is a typo error, and we have deleted it in L110 as "*Detailed model description of the WRF-Chem model, the calculation of aerosol optical properties, and activation of aerosols to CCN and IN can be found in Supplement Information (S1, S2, and S3).*".

26. L110: The time information is a little bit confused. Could you please revise it to "from UTC 1200 of July 21 to UTC 0000 of July 25, 2016" if this is not convention? Actually, I was a bit confused when I saw UTC 25 or so. Same for L118. You may keep it if you are sure this is conventional writing style.

**Response:** We appreciate your careful reading and agree that the time format could be clearer. We have revised the sentences in L132-133 and L145-146 to: "*The WRF-Chem is first integrated for an 84-h period from 1200 UTC (2000 LT) of July 21 to 0000 UTC (0800 LT) of July 25, 2016 for D01, with a 30-h spin-up time.*" and "*The WRF-Chem model is then integrated for a 24-h period from 0000 UTC (0800 LT) of July 24 to 0000 UTC (0800 LT) of July 25, 2016 for D02.*" We believe this revision addresses the confusion and presents the timing in a more conventional and understandable manner.

27. Figure 2: There is a strange symbol in the text of Figure 2. Also see Figure 5.

**Response:** We have revised the texts in Figure 2 as "*Pattern comparisons of simulated (color counters)* *versus observed (colored dots) near-surface mass concentrations of (a) PM$_{2.5}$, (b) O$_3$, (c) NO$_2$ and (d) SO$_2$ at 0000 UTC (0800 LT) on 24 July 2016.*", and in Figure 5 (Figure 7 in the revised versiin) as "*Pattern comparisons of simulated (color counters)* *versus observed (colored dots) accumulative precipitation on 24 July 2016.*"

28. Figure 7: Larger font size is preferred here.

**Response:** We have increased the font size in Figure 7 (Figure 9 in the revised version) as suggested.

[Figure]

**Figure 9: Average profile variation of (a) air temperature, (b) water vapor, and (c) vertical velocity in the GZB from 0000 to 0400 UTC (from 0800 to 1200 LT) caused by ARIs, as a function of the scale factor of anthropogenic emissions. The black line denotes the PBL height.**

29. L187: …, the deduction becomes more pronounced with increasing AESF, but is not sensitive to height.

**Response:** We have revised the sentence in L275-277 as "*The ARI effect lowers the temperature of the low-level atmosphere, and the temperature decrease becomes increasingly significant with increasing AESF, but is not sensitive to height.*"

---

## Author Comment (AC2)

**Reply to Anonymous Referee #2**

We thank the reviewer very much for the careful reading of our manuscript and helpful comments. We have revised the manuscript following the suggestions, as described below.

Comment to "Impacts of aerosol-radiation and aerosol-cloud interactions on a short-term heavy rainfall event - A case study in the Guanzhong Basin, China" by Bei et al.

This study investigates the impacts of aerosol-radiation and aerosol-cloud interactions on a short-term heavy rainfall event occurred in the Guanzhong Basin of central China using a cloud-resolving fully-coupled Weather Research and Forecasting model with Chemistry, with interesting findings. Particularly, the synergetic effect consistently decreases the precipitation in the whole domain with increasing aerosols, but ARIs play a more important role in the decreasing trend of the precipitation with deterioration of PM pollution. Personally, I think this study is worthy for publication with necessary modifications.

1. Line 48-53, Actually, it is also related to the relative location of aerosol and cloud vertical locations.

**Response:** We have revised the sentence in L55-56 as "*The impact of ACIs on precipitation varies under different meteorological conditions (Khain et al., 2008; Storer et al., 2010; Lebo and Morrison, 2014; Guo et al., 2016; Chen et al, 2020), cloud types (Tao, 2007; Lee et al., 2008), precipitation types (Guo et al., 2018; Sun and Zhao, 2021), cloud/precipitation development stages (Guo et al., 2014), aerosol composition and size distribution (Zhang et al., 2002; Jiang et al., 2018; Xi et al., 2024), the relative location of aerosol and cloud vertical locations (Ackerman et al., 2005; Sand et al., 2020; Senf et al., 2021), and orography conditions (Yang et al., 2014; Nugent et al., 2016).*"

*Sand, M., Samset, B. H., Tsigaridis, K., Bauer, S. E., and Myhre, G.: Black Carbon and Precipitation: An Energetics Perspective, J. Geophys. Res.-Atmos., 125, 10.1029/2019jd032239, 2020.*

*Senf, F., Quaas, J., and Tegen: Absorbing aerosol decreases cloud cover in cloud-resolving simulations over Germany, Q. J. R. Meteorol. Soc., 147, 4083-4100, 10.1002/qj.4169, 2021.*

2. Line 54-57, Supporting references should be provided, with Zhao et al. (2018, doi: 10.1002/2017EA000346) as suggested.

**Response:** We have added the supporting reference to the sentence in L60 as "*It has been well established that elevated aerosol concentrations increase the cloud droplet number concentration (CDNC), thus reducing cloud particle sizes, inhibiting collision and coalescence processes and increasing the cloud liquid (Zhao et al., 2018).*"

*Zhao, C., Qiu, Y., Dong, X., Wang, Z., Peng, Y., Li, B., Wu, Z., and Wang, Y.: Negative aerosol-cloud re relationship from aircraft observations over Hebei, China, Earth Space Sci., 5, 19-29, doi:10.1002/2017EA000346, 2018.*

3. Line 80-82, Why do previous studies focus on the mountain regions?

**Response:** We have added statements in L90-93 as "*Studies on the aerosol impact on precipitation in the GZB and surrounding areas (GZBs) are mostly focused on the rainfall over the mountain area due to the extensive long-term observational data available from Mt. Hua's summit. Additionally, the area is highly prone to orographic precipitation and is significantly influenced by aerosol transport from the heavily polluted upwind areas (Rosenfeld et al., 2007; Yang et al., 2013a; 2013b).*"

4. Line 87-90, Are there other similar studies regarding the synergetic effects of ARIs and ACIs over this region? If there are, a short introduction along with their findings are appreciated.

**Response:** We appreciate the reviewer's insightful query regarding regional studies on aerosol-radiation-cloud interactions. Following a systematic literature review, we confirm that peer-reviewed studies explicitly addressing the synergistic effects of ARIs and ACIs over the GZB remain exceptionally limited. All related studies in the region have been cited in our manuscript. This scarcity of regional studies further underscores the significance of our work in advancing process-level understanding of aerosol impacts on hydrological extremes in heavily polluted midlatitude environments.

5. Line 100-105, Why one-way instead of two-way nested grids are used?

**Response:** We used one-way nesting to strictly isolate aerosol effects (ARI/ACI) under controlled conditions. Two-way nesting would allow aerosol-induced changes in D02 to feedback to D01, altering meteorological initial/boundary conditions for D02. This would confound attribution of precipitation changes specifically to aerosol effects (ARI/ACI), as dynamical drivers (e.g., wind, moisture) would vary across simulations. One-way nesting ensures identical meteorological forcing (from D01) for all D02 sensitivity tests (e.g., F_BASE vs. F_ARI0), isolating aerosol impacts from external dynamical variability. We have added related statements in L117-121: "*The one-way nesting approach is intentionally adopted to prevent aerosol-induced changes in D02 from dynamically feeding back to D01, thereby maintaining identical meteorological forcing across all sensitivity experiments (e.g., F_BASE vs. F_ARI0). This isolation ensures that precipitation differences in D02 are solely attributable to aerosol effects (ARIs/ACIs) rather than confounding meteorological variability.*"

6. Line 110-114, How long for the spin-up?

**Response:** We have added spin-up time in the sentence in L133 as "*The WRF-Chem is first integrated for an 84-h period from 1200 UTC (2000 LT) of July 21 to 0000 UTC (0800 LT) of July 25, 2016 for D01, with a 30-h spin-up time.*"

7. Line 162-164, For these discrepancies, how do the authors explain them or how would they affect the study results?

**Response:** We have added discussions in L231-234 as "*These discrepancies primarily stem from uncertainties in meteorological field simulations (e.g., moisture transport, vertical wind shear), yet their consistent propagation across all sensitivity experiments minimizes impacts on aerosol effect quantification, as differences between simulations solely reflect aerosol perturbations.*"

8. Line 205, "Increased anthropogenic emissions …"

**Response:** Corrected.

9. Line 265-267, Actually, various studies have shown different results. While invigoration effects of clouds and enhancement of precipitation are found by many studies, a recent study has shown the vertical dependency of precipitation response to aerosols (Sun et al. 2023, doi: 10.1029/2022GL102186) – only precipitation relatively close to cloud bases are enhanced by the invigoration effect. In another word, evaporation effect could also play a role, which should be discussed.

**Response:** *We have added discussions in L419-422 as "Multifarious measurements and numerous modeling simulations have revealed that increased aerosols invigorate convective clouds and enhance precipitation (Cerveny and Balling, 1998; Shepherd and Burian, 2003; Khain et al., 2005; Lin et al., 2006; Tao, 2007; Li et al., 2008; Lee et al., 2018). Recent study has demonstrated aerosol-induced nonlinear regulation of convective precipitation-top heights via phase-change energy partitioning, showing invigoration-to-suppression transitions, with negligible near surface rainfall sensitivity due to boundary layer evaporation dominance (Sun et al., 2023)."*

*Sun, Y., Wang, Y., Zhao, C., Zhou, Y., Yang, Y., Yang, X., Fan, H., Zhao, X., and Yang, J.: Vertical dependency of aerosol impacts on local scale convective precipitation, Geophys. Res. Lett., 50, e2022GL102186, doi: 10.1029/2022GL102186, 2023.*